# How useful is snow accumulation in reconstructing surface air temperature in Antarctica? A study combining ice core records and climate models

Quentin Dalaiden[1], Hugues Goosse[1], François Klein[1], Jan T. M. Lenaerts[2], Max Holloway[3,4], Louise Sime[3], and Elizabeth R. Thomas[3]

[1]Georges Lemaître Centre for Earth and Climate Research (TECLIM), Earth and Life Institute (ELI), Université catholique de Louvain (UCL), Louvain-la-Neuve Belgium
[2]Department of Atmospheric and Oceanic Sciences, University of Colorado Boulder, Boulder CO, USA
[3]British Antarctic Survey, Madingley Road, Cambridge, CB3 0ET, UK
[4]Scottish Association for Marine Science, Oban, UK

**Correspondence:** Quentin Dalaiden (quentin.dalaiden@uclouvain.be)

**Abstract.** Improving our knowledge of the temporal and spatial variability of the Antarctic Ice Sheet (AIS) Surface Mass Balance (SMB) is crucial to reduce the uncertainties of past, present and future Antarctic contribution to sea level rise. An examination of the surface air temperature–SMB relationship in model simulations demonstrates a strong link between the two. Reconstructions based on ice cores display a weaker relationship, indicating a model-data discrepancy that may be due to model biases or to the non-climatic noise present in the records. We find that, on the regional scale, the modelled relationship between surface air temperature and SMB is often stronger than between temperature and $\delta^{18}$O. This suggests that SMB data can be used to reconstruct past surface air temperature. Using this finding, we assimilate isotope-enabled model SMB and $\delta^{18}$O output with ice-core observations, to generate a new surface air temperature reconstruction. Although an independent evaluation of the skill is difficult because of the short observational time series, this new reconstruction outperforms the previous reconstructions for the continental-mean temperature that were based on $\delta^{18}$O alone. The improvement is largest for the East Antarctic region, where the uncertainties are particularly large. Finally, using the same data assimilation method as for the surface air temperature reconstruction, we provide a spatial SMB reconstruction for the AIS over the last two centuries showing large variability in SMB trends at regional scale, with an increase (0.82 Gt year$^{-2}$) in West Antarctica over 1957–2000 and a decrease in East Antarctica during the same period (-0.13 Gt year$^{-2}$). As expected, this is consistent with the recent reconstruction used as a constraint in the data assimilation.

## 1 Introduction

The spatial coverage of climate observations in Antarctica and the Southern Ocean is sparse (e.g. Jones et al., 2016; Neukom et al., 2018). Consequently, the climate dynamics of the high southern latitudes are still poorly understood, leading to large uncertainties in the processes governing climate variability (Church et al., 2013). Since around 1995, the contribution to the global sea level rise from the ice sheets – Greenland Ice Sheet (GrIS) and the Antarctic Ice Sheet (AIS) – has strongly increased,

and are slowly outpacing the contributions from mountain glaciers and ocean thermal expansion (Shepherd et al., 2018). The GrIS has been dominating the ice sheet contribution so far (Rignot et al., 2019), but AIS mass loss has increased fivefold in 2012–2017 relative to 1992–1997, with current AIS mass loss values that approach those of the GrIS.

The (grounded) AIS Mass Balance (MB) is the difference between the surface mass balance (SMB) and the solid ice discharge (Lenaerts et al., 2019; Fyke et al., 2018). Reliable estimates of AIS MB and its relationship with internal climate variability and transient climate forcing are needed to constrain future climate and sea level projections (Bamber et al., 2018). The current AIS MB is negative (Rignot et al., 2019) because of large values of ice discharge (IMBIE team, 2018).

The SMB is defined as the difference between the incoming and outgoing mass at the surface of the ice sheet. In Antarctica, the main source term of the SMB, and its interannual variations, is precipitation in the form of snow (e.g. Lenaerts et al., 2012; Agosta et al., 2018). Unlike Greenland, AIS surface melt is small, and most surface melt water refreezes in place, not contributing to SMB (Trusel et al., 2015; Kuipers Munneke et al., 2012). As a result, the surface sublimation and sublimation of blowing snow are the main sink terms of the AIS SMB (e.g. Frezzotti et al., 2013; van Wessem et al., 2018).

Ice cores provide information on past changes in surface air temperature and SMB across Antarctica on time scales of centuries to millennia (e.g. Stenni et al., 2017; Thomas et al., 2017). In particular, it has become standard to reconstruct past temperature changes from water stable isotopes, and in particular $\delta^{18}$O (e.g. Jouzel, 2003; Masson-Delmotte et al., 2006). However, ice core studies suffer from several limitations: 1) the ice core network is still relatively sparse, despite recent coordinated international drilling efforts (Thomas et al., 2017; Stenni et al., 2017); 2) annually resolved surface air temperature and SMB records are not available from extremely dry areas, such as the East Antarctic Plateau; 3) changes in precipitation seasonality (e.g. Sime et al., 2008), moisture origin (e.g. Holloway et al., 2016a) and other processes can modify the expected relationship between $\delta^{18}$O and surface air temperature (e.g. Jouzel et al., 1997; Klein et al., 2019). Combined, these factors lead to large uncertainties in the reconstruction of surface air temperatures.

Until recently, AIS SMB had been considered to display no significant trends since the mid-twentieth century (Monaghan et al., 2006; Frezzotti et al., 2013). Based on recent work, this hypothesis has been revised: using a larger ice core network (PAGES2k database), Thomas et al. (2017) and Medley and Thomas (2019) have shown that AIS SMB has increased significantly since 1900, albeit with important regional differences. The Antarctic Peninsula has witnessed a considerable SMB increase during the twentieth century (e.g. Thomas et al., 2015; Goodwin et al., 2016), as well as some regions of Dronning Maud Land (e.g. Philippe et al., 2016; Lenaerts et al., 2013; Medley et al., 2018; Shepherd et al., 2012). In contrast, other regions of Droning Maud Land are subjected to a SMB decrease over the recent past (Schlosser et al., 2014; Altnau et al., 2015). All these studies point out the need to densify the ice-core network over Antarctica, but also to retrieve more insight in what is driving the trends in AIS SMB and its spatial signatures. For the latter, output of climate model simulations can be very useful (e.g Lenaerts et al., 2018).

In the last decade, output of several climate model simulations that cover the last millennium has become available (Schmidt et al., 2011). Thus far, model evaluation has been mainly focussed on surface air temperature (PAGES 2k-PMIP3 group, 2015). These results have shown discrepancies in AIS surface air temperature between climate model simulations and reconstructions during the last millennium. In contrast to climate model results, surface air temperature reconstructions show no clear warming

over the 20[th] century at the continental scale (Goosse et al., 2012; Stenni et al., 2017; PAGES 2k-PMIP3 group, 2015; Neukom et al., 2018). This mismatch can be explained by an overestimation of the response of climate models to external forcing, or the strong natural variability occurring in Antarctica or an underestimation of the signal in proxy-based reconstructions, or by a combination of all those (Jones et al., 2016; Neukom et al., 2018). Unlike temperature changes, modelled AIS SMB variations over the past millennium are poorly documented.

In a warmer climate, AIS SMB is expected to increase due to higher snowfall associated to the greater moisture holding capacity at higher air temperature (e.g. Lenaerts et al., 2016). Taken alone, this straightforward thermodynamical effect would mitigate the sea level rise (Huybrechts et al., 2004; Krinner et al., 2007; Frieler et al., 2015). According to Frieler et al. (2015), the observed sensitivity of Antarctic snowfall accumulation to surface air temperature was about 5% K$^{-1}$ during the 1960–1999 period. Based on climate model simulations, this sensitivity is expected to increase in future with an estimated conversion value of 7.4% K$^{-1}$ for the end of the 21st century (2080–2099; Palerme et al., 2017). The link between surface air temperature and SMB has been confirmed for small regions at the centennial time scale (200 years; e.g. Oerter et al., 2000; Medley et al., 2018) and on longer time scales (glacial-interglacial; Frieler et al., 2015) for the full AIS using climate models and ice cores. However, some studies using surface air temperature reconstructions based on $\delta^{18}$O data (Fudge et al., 2016; Altnau et al., 2015; Philippe et al., 2016; Goursaud et al., 2019) suggest that this SMB-surface air temperature relationship is not always positive and varies spatially and temporally. These results suggest that in some regions, especially along the AIS coasts, the SMB variability is dominated by large-scale atmospheric circulation rather than by thermodynamic processes (such as the Clausius-Clapeyron relation), limiting the correlation with $\delta^{18}$O.

The first goal of this study is to document the relationship between surface air temperature and SMB in Antarctica on a regional scale using climate models and ice-core records over the two past centuries and over the last millennium. The final goal is to use the covariance between both variables to reconstruct past changes over the last two centuries by using a data assimilation procedure. All reconstruction methods depend on the number and quality of the input data. However, while the statistical methods classically used to infer past surface air temperature (see for instance Stenni et al., 2017) rely on the length of the calibration period, on the quality of the record during this period, and on the stationarity of the link between the proxy and the variable of interest, which can be strong assumptions in the case of the $\delta^{18}$O-temperature relationship (Klein et al., 2019), data assimilation does not. In recent years, data assimilation has become a standard procedure in paleoclimatology to optimally combine the information from model results and proxies and to provide estimates of past climate states (e.g. Hakim et al., 2016; Widmann et al., 2010; Goosse et al., 2010; Matsikaris et al., 2015; Steiger et al., 2014). Nevertheless, Antarctic SMB to the best of our knowledge has never been assimilated in a climate model. The biggest advantage of using data assimilation is that it takes into account information brought by both SMB and $\delta^{18}$O without making the strong assumptions that the statistical methods do. Additionally, using the covariance between them might lead to better estimates of past changes in the two variables, particularly when proxy records are scarce and few instrumental data are available, which is the case for the Antarctica. The resulting reconstructions will have the benefit of being compatible with the physics of the climate system as represented by the models.

## 2 Data: model simulations and observations

### 2.1 Global climate model simulations

The climate model simulations selected for this study are those for which the required variables (i.e. precipitation and sublimation/evaporation) are available for the last millennium from the PMIP3-CMIP5 database (Otto-Bliesner et al., 2009; Taylor et al., 2012). In addition to these simulations, the CESM1-CAM5 model simulations covering the last millennium (Otto-Bliesner et al., 2015) are also used. The characteristics and references of each model are described in Tab. A1. All these GCMs use the GMTED2010 elevation dataset (Danielson and Gesch, 2011) as topography, adapted to their spatial horizontal resolution. The simulations are driven by both natural (orbital, solar and volcanic) and anthropogenic (greenhouse gases, land use, aerosol and ozone) forcings through the last millennium (Schmidt et al., 2011, 2012). Except for CESM1-CAM5, CSIRO-Mk3L-1-2 and MPI-ESM-P, the simulations do not cover the entire millennium. Historical simulations covering 1851-2005 CE were launched independently of simulations covering 850-1850 CE (referred to as past1000 experiment). In order to obtain results over the full millennium, we adopt the approach from Klein and Goosse (2018) and merge the first ensemble members (r1i1p1) of the past1000 experiment with the corresponding ensemble members of the historical experiment. Although not continuous, there is no large discrepancy between the two merged simulations (e.g. Klein and Goosse, 2018).

Simulations performed with the isotope-enabled climate models, ECHAM5-MPI/OM (Sjolte et al., 2018), ECHAM5-wiso (Steiger et al., 2017) and iHadCM3 (Tindall et al., 2009; Holloway et al., 2016b) are also analyzed. These simulations allow for a direct comparison with observed water isotope content. ECHAM5/MPI-OM is a fully coupled General Circulation Model (GCM). The simulation used here covers the period 800–2000 CE forced by natural and anthropogenic forcing (Sjolte et al., 2018). The horizontal resolution of the atmospheric model is $3.75° \times 3.75°$. The simulation of ECHAM5-wiso, which only includes an atmospheric component, was performed by Steiger et al. (2017) and covers the period 1871–2011 CE at $\sim 1°$ resolution. The model is driven by the sea surface temperature and sea ice from the Rayner et al. (2003) dataset. Due to a lack of Antarctic sea ice data before 1973, this dataset is based on historical climatologies of sea ice concentration for the period 1871–1973 CE, with no interannual variability. Finally, iHadCM3 is the version of HadCM3 (fully coupled climate model; Turner et al., 2016) which has an explicit representation of the water isotopes. The resolution of the atmospheric model is $3.75° \times 2.5°$. While only one simulation is available for ECHAM5-MPI/OM and ECHAM5-wiso, we have an ensemble of seven iHadCM3 simulations spanning the industrial period from 1851 to 2003 CE. The initial conditions for each of these simulations correspond to different years in the pre-industrial control simulation of the iHadCM3 model. Comparisons of the results of these three isotope-enabled models with modern $\delta^{18}O$ observations indicate that they all reproduce the main characteristics of the spatial distribution of the isotopic composition of precipitation over Antarctica including the latitudinal distribution (negative $\delta^{18}O$ gradient from the coasts to the Plateau). According to Tindall et al. (2009) and Sime et al. (2008), the small biases in $\delta^{18}O$ (for example, an underestimation of the spatial $\delta^{18}O$ variability in rugged areas) in the iHadCM3 simulation mainly come from the coarse horizontal resolution of the model and not from the isotopic model itself. ECHAM5-wiso and ECHAM5/MPI-OM display an overall underestimation of $\delta^{18}O$ in Antarctica but reproduce well the general Antarctic $\delta^{18}O$ pattern (Goursaud et al., 2018; Klein et al., 2019, see reference of each model for more details).

Klein et al. (2019) has recently described an evaluation of Antarctic surface air temperature in reconstructions and model simulations over the last millennium. In accordance with Abram et al. (2016), they highlighted the early onset of industrial warming simulated by the PMIP/CMIP models, which is not observed in the $\delta^{18}$O-based temperature reconstructions of Stenni et al. (2017). This suggests that the Antarctic surface air temperatures simulated by the models are too sensitive to the anthropogenic forcing.

## 2.2 The regional climate model RACMO2 simulation

The evaluation of AIS SMB simulated by GCMs for the present period (1979–2005) is mainly based on the results of the regional atmospheric climate model RACMO2.3p_2 (RACMO2 hereafter) covering the entire AIS over 1979–2016 (van Wessem et al., 2018). This is because 1) the SMB observations are very sparse on the AIS (Favier et al., 2013); 2) the interannual (year-to-year) variability is different between observations and GCMs given that the latter are freely-evolving coupled models. Consequently, the comparison can be only made on multi-decadal time scales (> 20 years), which drastically reduces the availability of observations; 3) unlike observations, RACMO2 provides a complete SMB field over the entire AIS; and, finally, (4) RACMO2 has been extensively evaluated against available measurements and displays a very good agreement (e.g. van Wessem et al., 2018; Lenaerts et al., 2012). In an intercomparison of AIS SMB from reanalysis, atmospheric models and observations, Wang et al. (2016) showed that the RACMO2 model best fits the recent AIS SMB observations compared to all other available datasets.

RACMO2 combines the physics package of the European Centre for Medium- Range Weather Forecasts (ECMWF, 2008) integrated Forecast System and the hydrostatic dynamics of the High Resolution Limited Area Model (HIRLAM, Unden et al., 2002). RACMO2 is specially adapted to polar regions since it includes the interactions between the atmosphere and the multi-layered snow model that calculates physical processes occurring in the firn: meltwater production, percolation, runoff, refreezing, as well as snow grain size and resulting snow albedo (Greuell and Thomas, 1994; Ettema et al., 2010). RACMO2 also includes a drifting snow scheme simulating the interactions between the near-surface air with snow (Lenaerts et al., 2010). All the SMB components are explicitly calculated by this regional model on a 27 km resolution grid. The Digital Elevation Model of Bamber et al. (2009) is taken as reference of the Antarctic topography. ERA-Interim reanalysis data (Dee et al., 2011) are used to force the regional model at its lateral boundaries. For more details on RACMO2, see van Wessem et al. (2018).

## 2.3 Snow accumulation database from Antarctica2k

The annually resolved Antarctica2k (Ant2k) snow accumulation database (Thomas et al., 2017) is used for the evaluation of AIS SMB simulated by GCMs before 1979. The estimate of the SMB from ice cores is based on the physical distance between suitable age markers within the ice core. The age markers used depend on the timescale of interest ranging from glacial cycles (e.g. bulk changes in isotopic compositions) to seasonal variations reflected by changes in stable water isotopes, while volcanic eruptions can inform on decadal to millennial timescales (Dansgaard and Johnsen, 1969). Once the age markers are identified, since the firn density generally increases with depth in the ice core, it is necessary to consider those variations to convert the age and depth to mass (Van Den Broeke et al., 2008). Doing so, SMB is converted to meters of water equivalent based on

measured density and corrected for the vertical strain rate effect – the differential vertical velocity with depth leading to layer thinning with depth (Thomas et al., 2017).

This database is composed of 79 records that are assigned to seven geographical regions (Fig. 1) with distinctly different climates. East Antarctica above 2000 m elevation constitutes the East Antarctica Plateau (EAP). West Antarctica is separated into two parts: the Antarctic Peninsula (AP) and the West Antarctica Ice Sheet (WAIS), with a division at 88° W. The coastal region of East Antarctica is divided into four regions: Victoria Land (VL; 150-170° E), the Wilkes Land Coast (WL; 70-150° E), Dronning Maud Land (DML; 15° W-150° E) and the Weddell Sea Coast (WS; 15-60° E). For each region, this database covers the past 1000 years except for EAP, AP and DML, for which the period covered is 1240–2005 CE, 1703–2010 CE and 1737–2010 CE, respectively. Hereafter, West Antarctica is composed of WAIS and AP, while East Antarctica comprises all of the other regions. Since some Antarctic regions lack long-term data, the SMB reconstruction for the whole Antarctic ice sheet is only available from 1737 AD. This regional SMB reconstruction has been compared to RACMO2, concluding that the reconstruction captures a large proportion of the regional spatial SMB variability as defined by RACMO2 for the 1979–2010 period (Thomas et al., 2017).

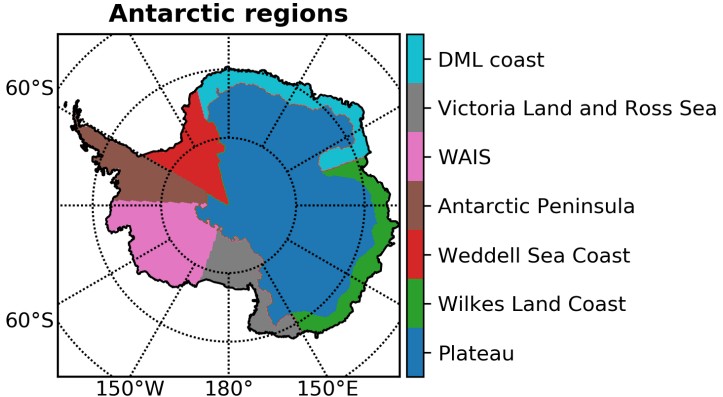

**Figure 1.** Antarctic regions used in this study. The definitions of the regions are those of Thomas et al. (2017).

## 2.4 Water stable isotopes records and surface air temperatures reconstructions from Antarctica2k

Stenni et al. (2017) built $\delta^{18}$O regional composites from 112 individual ice cores compiled in the framework of the PAGES Antarctica2k working group for similar seven Antarctic subregions as in Thomas et al. (2017; see Sec. 2.3) over the last two millennia. Based on those $\delta^{18}$O composites, they reconstructed regional surface air temperatures over the last two millennia based on the statistical relationship between $\delta^{18}$O and surface air temperature. Three methods have been used to scale the $\delta^{18}$O composites. The second reconstruction (*borehole* reconstruction) is used throughout this study for two reasons: 1) this is not based on surface air temperature observations, which are used here to estimate the skill of the reconstructions which would have led to a bias; 2) because it is based on more information, the borehole reconstruction is expected to be more accurate (see

section S1 for details). The temporal resolution is the same as for the $\delta^{18}$O composites: 10 years over 0–1800 and 5 years over 1800–2010.

# 3 Methods: Reconstructing SMB and surface air temperatures using data assimilation

## 3.1 Data assimilation method: a particle filter using fixed ensembles

Data assimilation optimally combines observations (proxy data in our case) and climate model states. Two types of data assimilation methods are usually applied in paleoclimatology. First, online methods follow standard sequential data assimilation approaches, in which the analysis at a single time step depends on the state at the previous step. Information is thus propagated forward in time. However, because data assimilation requires a large ensemble of model simulations (tens to hundreds), for paleoclimate reconstructions, performing online data assimilation at high spatial climate model resolution (e.g. CMIP5 class

as used here) becomes impractical. Second, when working with so-called offline methods, ensemble members are constructed from existing model simulations, which is of great interest in terms of computation time compared to online methods. Here, ensemble members are constructed by individual years and not by independent model simulations. Therefore, in contrast to online methods, offline methods do not maintain temporal consistency. However, when the predictability on inter-annual time-scales is limited, such as surface air temperature or precipitation because of the dominant role of their chaotic nature,

online methods do not outperform offline ones (Matsikaris et al., 2015). Indeed, offline methods have provided skilful data assimilation-based reconstructions for various types of data (e.g Steiger et al., 2017; Klein and Goosse, 2018; Hakim et al., 2016). Nevertheless, the online approach is preferred when focussing on ocean dynamics because of the ocean long memory (e.g. Goosse, 2017; Pendergrass et al., 2012).

The offline data assimilation method applied in this study is based on a particle filter (e.g. van Leeuwen, 2009; Dubinkina

et al., 2011) using fixed ensembles from climate model outputs. The implementation described in Dubinkina et al. (2011) is identical to previous studies (e.g. Klein and Goosse, 2018). Hence, only a brief description of the methodology will be given here. At each time step of the data assimilation procedure (yearly, see Sec. 3.2), each ensemble member, called particle, is compared to the proxy-based reconstruction by computing its likelihood, assumed here to be Gaussian, taking into account data uncertainties (see Dubinkina et al. (2011) for details). Depending on its likelihood, each particle receives a weight. Then,

all the weights are multiplied by the number of particles and rounded to the nearest integer toward negative infinity by ensuring that the sum of the weights equals the number of particles throughout the data assimilation process (see Dubinkina et al. (2011) for details). Considering all particles weights, we can compute a weighted average, providing a reconstruction for this time step. In this study, the ensemble members are derived from the three isotope-enabled climate model outputs ECHAM5-MPI/OM, ECHAM5-wiso and iHadCM3, presented at Section 2.1. These models have been chosen because they explicitly simulate

$\delta^{18}$O. Because iHadCM3 offers an ensemble of seven simulations, while the other isotope-enable models have only a single realization, we mainly focus on the iHadCM3 outputs in the main manuscript. The results from the other models are shown in the Supplementary Materials and provide similar conclusions to the ones obtained with iHadCM3. Using an ensemble allow us estimating the contribution of internal variability over the last century and the range provided by this ensemble is larger than the

one given by the other two models, increasing the probability of finding model results close to the assimilated records during the data assimilation process. Additionally, iHadCM3 ensemble provides a more realistic Antarctic surface air temperature and snow accumulation over recent past than the ECHAM5-MPI/OM simulation (Fig. S8; Klein et al., 2019).

## 3.2 Experiment design

Data assimilation is used in this study to reconstruct surface air temperature and SMB by taking advantage of the covariance between these variables. They are assimilated together as well as separately in three different experiments. In the first experiment, the seven subregion composites of $\delta^{18}O$ data (Stenni et al., 2017) are used to constrain model results. Assimilating $\delta^{18}O$ instead of surface air temperature potentially accounts for the non-stationary and the non-linearity of the stable oxygen ratios–surface air temperature link (Masson-Delmotte et al., 2008; Klein et al., 2019). For the second experiment, the SMB reconstruction for the seven subregions (Thomas et al., 2017) is used in the data assimilation process. Finally, both $\delta^{18}O$ and SMB are taken into account together in the last experiment. This allows us to estimate independently the consistency of the SMB and surface air temperature reconstructed between the various records and model results. In addition, our experiments allow us to assess the information acquired on surface air temperature by assimilating SMB, and on SMB by assimilating $\delta^{18}O$. In all the experiments, we assimilate annual-mean proxies. All modelled $\delta^{18}O$ are precipitation-weighted as this quantity is most similar to the one measured in ice cores.

Since the amount of ice cores is limited before 1800 CE (both for $\delta^{18}O$ and for SMB), which drastically decreases the quality of the regional composites (Thomas et al., 2017), the experiments are performed on the 1800–2010 period. Contrary to the SMB composites, which have an annual resolution, the composites of $\delta^{18}O$ are 5-year averages. Consequently, the $\delta^{18}O$ data have been interpolated linearly over the studied period to match the temporal resolution of the SMB reconstruction. However, as recommended by Stenni et al. (2017), the results are analyzed only for the 5-year averages. This temporal averaging reduces uncertainties in dating and the noise induced by non-climatic processes (e.g. Laepple et al., 2018; Fan et al., 2014).

In order to assess the skill our data assimilation-based surface air temperature reconstructions, we evaluate them at first with the reconstructions of Stenni et al. (2017). But this is biased since they are only based on $\delta^{18}O$ and we cannot thus evaluate the added value brought by SMB data and model physics in the data assimilation experiments. Therefore, independent data is needed to properly assess the potential of SMB and $\delta^{18}O$ in reconstructing surface air temperature. This is done here using the surface air temperature reconstruction from Nicolas and Bromwich (2014), which is based on surface air temperature records and not on $\delta^{18}O$ data, over the 1958–2010 period. SMB estimates are also available for the last decades (e.g. Medley et al., 2014), but they cover a too short period or have a too small spatial coverage to provide an independent validation of our reconstruction. It is thus not possible to estimate if the assimilation of SMB and $\delta^{18}O$ measurements provides an improvement for this field.

## 4   Results

### 4.1   Reconstructed and simulated SMB changes over the last centuries

The AIS SMB over the last millennium has been estimated for each GCM by computing the difference between precipitation and sublimation/evaporation. Runoff is assumed to be negligible as surface meltwater generally refreezes in the cold firn (Magand et al., 2008; Kuipers Munneke et al., 2012). Our short evaluation of SMB simulated by GCMs over the present-day (see section S3) suggests that the selected GCMs (including the isotope-enable models) display reasonable SMB climatology when compared to RACMO outputs.

Before the 19[th]th century, all GCMs simulations are characterized by large decadal variability, but no long-term trend (Fig. 2). A positive trend, albeit initiated at different times, is shown at the end of the simulation (around 1950 AD). All models agree on an AIS SMB increase from ∼1975 onwards, which is consistent with the SMB reconstruction of Thomas et al. (2017). However, the contrast in the SMB trends between East Antarctica and West Antarctica is clearly stronger in the reconstruction based on ice cores than in GCMs on average. Indeed, over the last decades (1950–2000), the ice core SMB reconstruction shows a large increase for West Antarctica (25.6 Gt year$^{-1}$ per decade) and a small decrease (-3.6 Gt year$^{-1}$ per decade) for East Antarctica, while, on average, the GCMs simulate a strong SMB increase in both regions (8.9 $\pm$ 9.2 Gt year$^{-1}$ per decade and 14.2 $\pm$ 13.5 Gt year$^{-1}$ per decade respectively; Figs. 2 and 3 and Tab. S1). When analyzing the ensemble of simulations performed with CESM1-CAM5, the ensemble mean also shows a relatively homogeneous increase, but some simulations display a contrast between East Antarctica and West Antarctica close to the one observed in the reconstruction (Fig. 3). This suggests that internal variability has a dominant contribution in the current Antarctic SMB changes and might explain why the observed contrast between East and West Antarctica is only present in a few simulations.

### 4.2   Relationship between SMB and surface air temperatures in Antarctica

Averaged across all GCMs, the relationship between SMB and surface air temperature is positive for each Antarctic region (Fig. 4). A very similar result is obtained when the annual mean surface air temperature and SMB derived from the RACMO2 simulation over the recent period (1979–2016) are used. The regional correlations are much weaker for the reconstructions based on ice cores than those obtained from model outputs (Fig. 4). These results are also true for detrended times series, indicating that this modelled link is valid at the inter-annual time-scale (not shown).

To quantify more precisely the link between surface air temperature and SMB in model outputs and reconstructions, the SMB sensitivity to temperature – defined as the slope of the linear fit between surface air temperature (SAT) and SMB – has been calculated. Firstly, the GCMs and reconstruction (i.e. Thomas et al., 2017; Nicolas and Bromwich, 2014) suggest that this sensitivity is similar for both West Antarctica and East Antarctica over the 1950–2000 period (Fig. 3). Secondly, on average over the entire continent, this sensitivity reaches 3.6 % K$^{-1}$ in ice cores-based reconstructions for the 1850–1949 period. According to these reconstructions, this sensitivity has increased a lot for the recent period (1950–2005; 15.52 % K$^{-1}$), confirming the findings of Frieler et al. (2015). However, Frieler et al. (2015) do not obtain such an increase in SMB sensitivity (only ∼+40%). Additionally, this recent increase found here in the reconstructions is not represented by the GCMs:

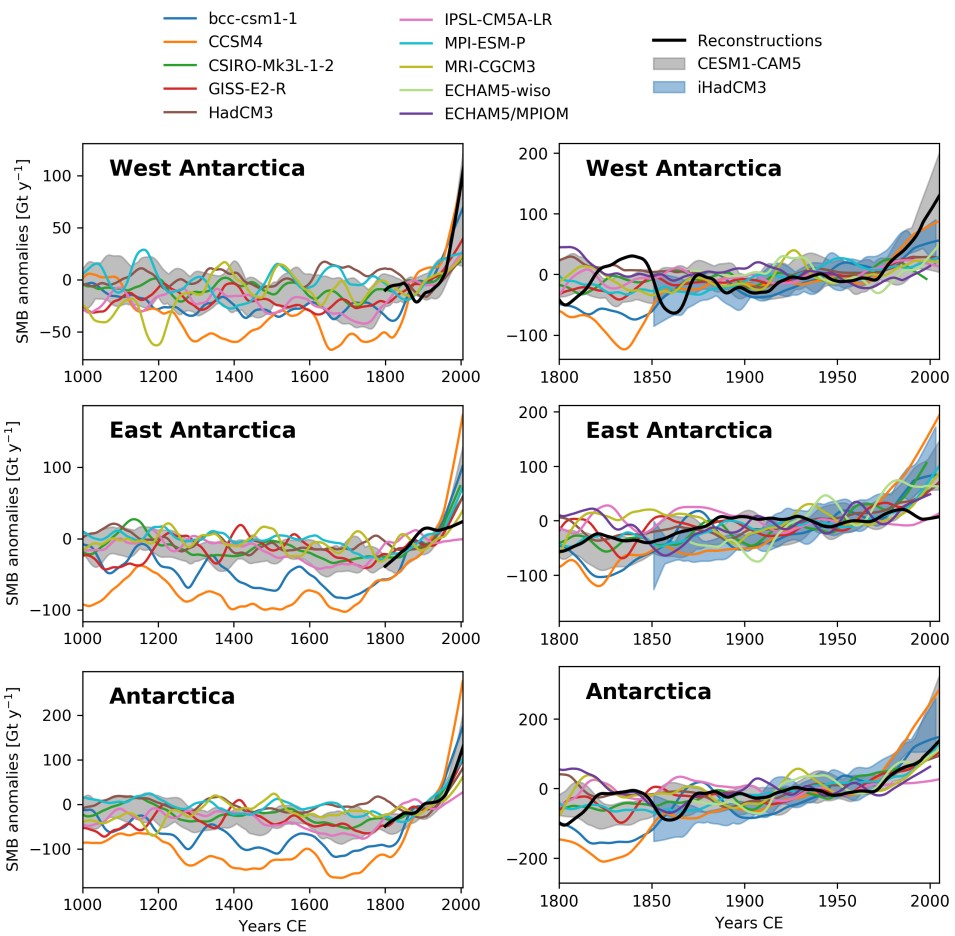

**Figure 2.** Surface Mass Balance anomalies [Gt y⁻¹] simulated by the GCMs (Tab. A1) and snow accumulation reconstructions (Thomas et al., 2017) during 1000 to 2005 and during 1800 to 2005 for West Antarctica, East Antarctica and Antarctica as a whole. Anomalies are relative to the 1871–2000 period. The shaded area corresponds to the range of the CESM1-CAM5 simulations. For visibility, data has been smoothed with a 100 years moving average for the last millennium and a 30 year moving average for the last 200 years. The equivalent for the seven subregions is given on Fig. S1.

on average, the simulated sensitivity of SMB to surface air temperature is 5.0 ± 1.1 % K⁻¹ over 1850–1949 and 5.4 ± 2.0 % K⁻¹ over 1950–2005. When looking at the regional scale over 1850–2005, the average SMB sensitivity over all models for West Antarctica (6.8 % K⁻¹) is in good agreement with the one deduced from the reconstructions (8.0 % K⁻¹; Fig. 5), while for East Antarctica, the sensitivity of the model mean is higher than the one obtained from the reconstructions (6.2 % K⁻¹ and 2.1 % K⁻¹ respectively). The very low SMB sensitivity in the reconstructions for East Antarctica, especially on the Antarctic Plateau (0.5 % K⁻¹) is somewhat unexpected, given that this region is continental and thus less affected by synoptic activities than coastal areas (Monaghan and Bromwich, 2008). Actually, when using the observed surface air temperatures (e.g. Nicolas

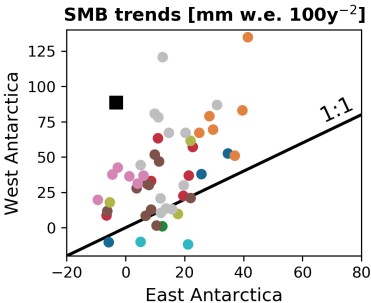
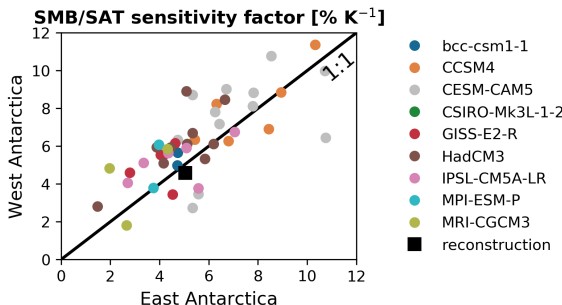

**Figure 3.** (left) Comparison between the reconstructed and the simulated SMB trends (mm w.e./100y$^{-2}$) over the period 1950–2000 CE in West Antarctica (y axis) and East Antarctica (x axis). (right) As on the left but for SMB/SAT sensitivity factors (% K$^{-1}$). For the reconstruction, data from Thomas et al. (2017) and Nicolas and Bromwich (2014) are used.

and Bromwich, 2014) instead of the reconstructed ones of Stenni et al. (2017), the Antarctic SMB sensitivity to temperature is strongly reduced (4.02 % K$^{-1}$ for the 1958–2010 period), and thus closer to the resulting sensitivity found in the GCMs (5.4 $\pm$ 2.0 % K$^{-1}$ for the 1950–2005 period).

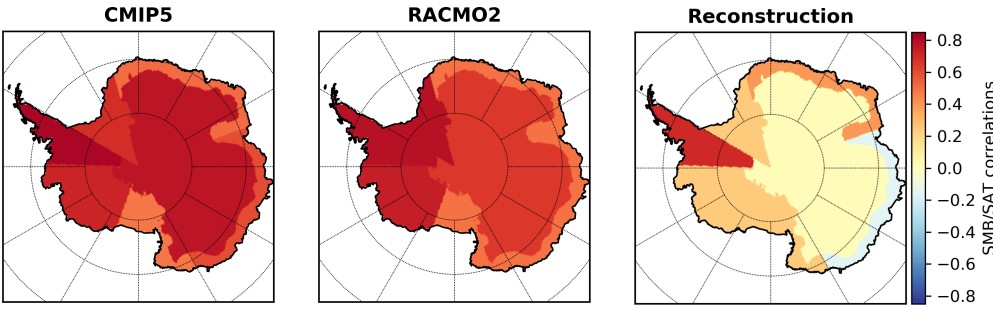

**Figure 4.** 5 yearly correlations (r) between SMB and surface air temperature for seven Antarctic regions (see Fig. 1 for geographical definitions) for GCMs over the 1850–2000 CE (left), for RACMO2 over 1979–2016 CE (center) and for ice core reconstructions (Thomas et al., 2017; Stenni et al., 2017) for 1850–2000 CE (right). For the CMIP5 models and RACMO2, their correlations are all statistically significant (p-value<0.05). For the reconstructions, the statistically significant (p-value<0.05) correlations are obtained for the Antarctic Peninsula and Dronning Maud Land Coast. See Fig. S2 for the correlations for each CMIP5 model.

    In the study of Neukom et al. (2018), the authors argue that the data sampling, the noise in proxy data and the deficiencies in the reconstruction methods can partly explain the discrepancy between models and reconstructions for the surface air temperature during the last millennium, especially for the southern hemisphere. The spatial coverage of the surface air temperature and SMB reconstructions based on ice cores is poor, in particular for East Antarctica (Stenni et al., 2017; Thomas et al., 2017). Moreover, due to the low snow accumulation in some regions, the uncertainties of the reconstruction are large for both surface

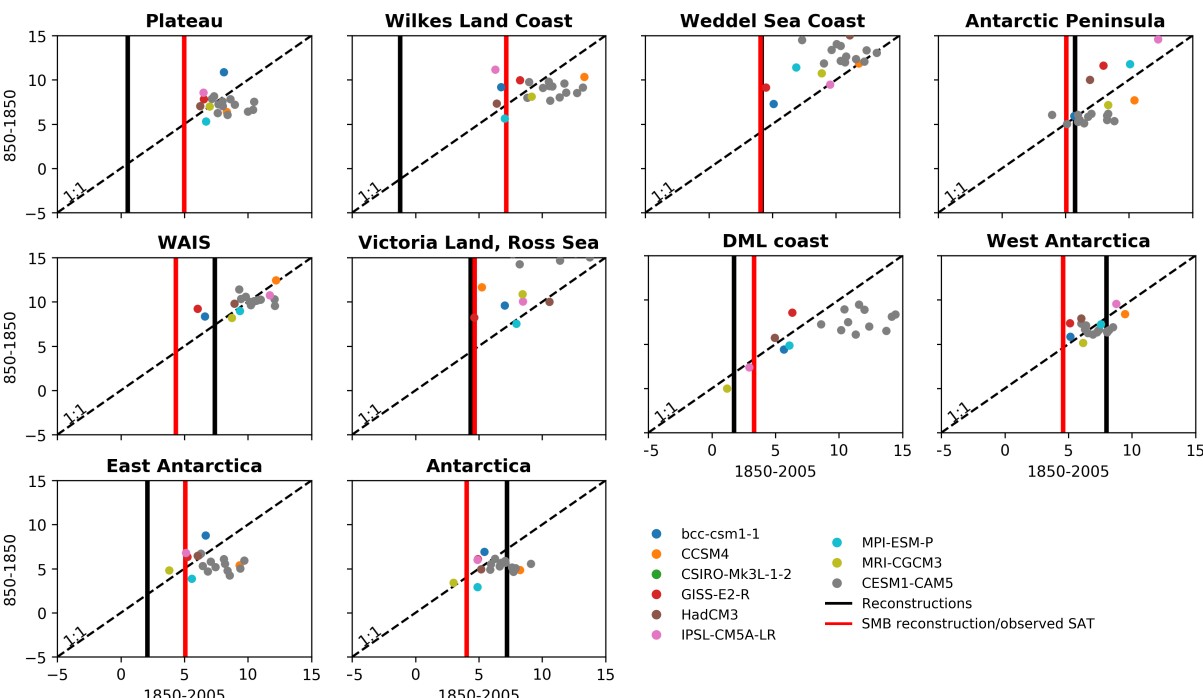

**Figure 5.** SMB sensitivity to Surface Air Temperature (SAT) over the 850–1850 and the 1850–2005 periods for each Antarctic region (see Fig. 1 for geographical definitions) for GCM outputs. Additionally, the SMB-$\delta^{18}$O sensitivity for ice cores based-reconstructions (i.e Thomas et al., 2017; Stenni et al., 2017) over 1850–2005 is represented by a solid black vertical line while a solid red vertical line represents the SMB-observed surface air temperature sensitivity (i.e. Thomas et al., 2017; Nicolas and Bromwich, 2014) over 1960–2010. For the CESM1-CAM5 model, the 12 simulations are plotted as grey points.

air temperatures and SMB, leading to noise in the time series (Stenni et al., 2017; Thomas et al., 2017; Frezzotti et al., 2007). Since the SMB reconstruction is only based on direct snow accumulation measurements, this is expected to be more accurate than the $\delta^{18}$O-based temperature reconstruction, which is built by assuming a stationary link between $\delta^{18}$O and surface air temperature. Because a lot of processes (such as precipitation seasonality or moisture origin) can significantly modify this

5 relationship over time (e.g. Jouzel et al., 1997; Sime et al., 2008), this is computed over a short calibration period, but this might be too short to be representative (Klein et al., 2019). Consequently, all these processes could explain the large differences between models and proxy-based reconstructions in the estimation of SMB sensitivity to surface air temperatures.

## 4.3 SMB and surface air temperature reconstructions from data assimilation

The high correlation values obtained between SMB and surface air temperatures in GCMs suggest that we can potentially use SMB to reconstruct Antarctic surface air temperature. The analysis of isotope-enabled model results reinforces this hypothesis (Fig. 6): the iHadCM3 outputs show high correlations between these two variables. For most regions, the link between surface air temperature and SMB (r=0.70 on average over the seven subregions for the 1850–2000 period) is higher than that between surface air temperatures and $\delta^{18}$O (r=0.55 on average over the seven subregions for the 1850–2000 period). This is consistent with the observations: the regional correlations between SMB from ice cores (e.g. Thomas et al., 2017) and the observed surface air temperatures (i.e. Nicolas and Bromwich, 2014) are high for several regions over the 1960–2010 period (using 5-year averages as for Stenni et al., 2017). In particular, this correlation for East Antarctica is 0.82 (statistically significant). The results with the outputs of ECHAM5-wiso and ECHAM5/MPI-OM are a bit more nuanced than those from iHadCM3 (Fig. S3). The results of ECHAM5-wiso and ECHAM5/MPI-OM confirm this strong link between SMB and temperature but, in contrast to iHadCM3, the correlations are not systematically higher than between $\delta^{18}$O and temperature. When analyzing the long ECHAM5/MPI-OM simulation (800–2000), the relationship between SMB and surface air temperature is generally higher than between $\delta^{18}$O and surface air temperature but the difference is small. For some regions, the SMB-surface air temperature link is much higher than the $\delta^{18}$O-surface air temperature link but it is weaker for other regions. Compared to the $\delta^{18}$O-surface air temperature link, the SMB-surface air temperature is also less spatially variable (minimum regional correlation is 0.54 against 0.07 for the $\delta^{18}$O-surface air temperature link).

### 4.3.1 Surface air temperatures reconstruction

When constraining the model with the SMB reconstruction of Thomas et al. (2017), the obtained surface air temperature reconstruction is less well correlated with the reconstruction of Stenni et al. (2017) than for the data assimilation reconstruction constrained by only the $\delta^{18}$O (Fig. 7). However, the difference is relatively small, despite the fact that SMB and surface air temperatures are more strongly correlated in models than in the ice core reconstruction (0.86 for iHadCM3 against 0.16 for ice cores; Fig. 6). When compared to observed surface air temperature over the 1958–2010 period (i.e. Nicolas and Bromwich, 2014), the surface air temperature reconstruction of Stenni et al. (2017) as well as the reconstruction when only $\delta^{18}$O is assimilated is in good agreement with the observed surface air temperatures for West Antarctica (Tab. 1, coefficient correlations are 0.79 and 0.69 respectively, both statistically significant) but not for East Antarctica (coefficient correlations are 0.10 and 0.13 respectively, both not statistically significant).

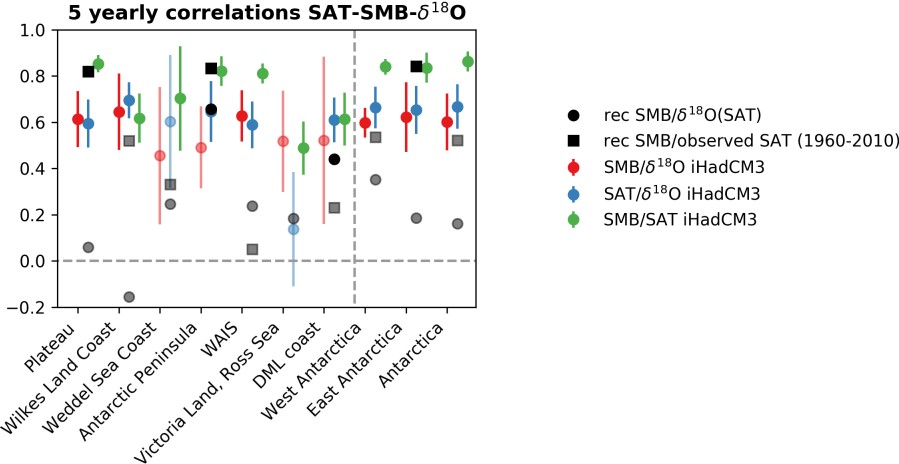

**Figure 6.** 5 year correlations between SMB and $\delta^{18}$O, surface air temperature and $\delta^{18}$O, and SMB and surface air temperature for the seven Antarctic regions over 1850–1995 period from the iHadCM3 outputs. The error bars correspond to the range (maximum and minimum) of the iHadCM3 simulations while the dot is the mean of the simulation ensemble. In black circles, the correlation between the SMB ice core reconstructions from Thomas et al. (2017) and the $\delta^{18}$O of Antarctic ice cores aggregated for the seven Antarctic regions (Stenni et al., 2017). In black squares, the correlation between the SMB reconstructions from Thomas et al. (2017) and the observed surface air temperatures aggregated for the seven Antarctic regions (Nicolas and Bromwich, 2014). This latter dataset covers only the 1960–2010 period (5-year averages). Non-significant correlations (p-value>=0.05) are shown in pale.

**Table 1.** 5-year mean correlations between the three surface air temperature reconstructions from data assimilation experiments using the ECHAM5-MPI/OM, ECHAM5-wiso and iHadCM3 outputs and the statistical reconstruction of Stenni et al. (2017), with the surface air temperature reconstructions from Nicolas and Bromwich (2014) over the 1958–2010 period for West Antarctica, East Antarctica and Antarctica as a whole. Stars represent statistically significant correlations (p-value<0.10). DA stands for data assimilation.

| | West Antarctica | | | East Antarctica | | | Antarctica | | |
|---|---|---|---|---|---|---|---|---|---|
| | ECHAM5-MPI/OM | ECHAM5-wiso | iHadCM3 | ECHAM5-MPI/OM | ECHAM5-wiso | iHadCM3 | ECHAM5-MPI/OM | ECHAM5-wiso | iHadCM3 |
| DA $\delta^{18}$O | 0.57* | 0.78* | 0.69* | 0.19 | 0.08 | 0.13 | 0.50 | 0.47 | 0.34 |
| DA SMB | 0.40 | 0.52 | 0.55 | 0.27 | 0.53 | 0.60* | 0.28 | 0.58* | 0.65* |
| DA $\delta^{18}$O and SMB | 0.53 | 0.65* | 0.72* | 0.34 | 0.48 | 0.61* | 0.59* | 0.71* | 0.73* |
| Stenni et al. (2017) | 0.79* | | | 0.10 | | | 0.57* | | |

In contrast to the data assimilation experiment, in which only $\delta^{18}$O is assimilated, the skill of the surface air temperature reconstruction is almost identical for both regions in the data assimilation experiment where only SMB is assimilated: r=0.55

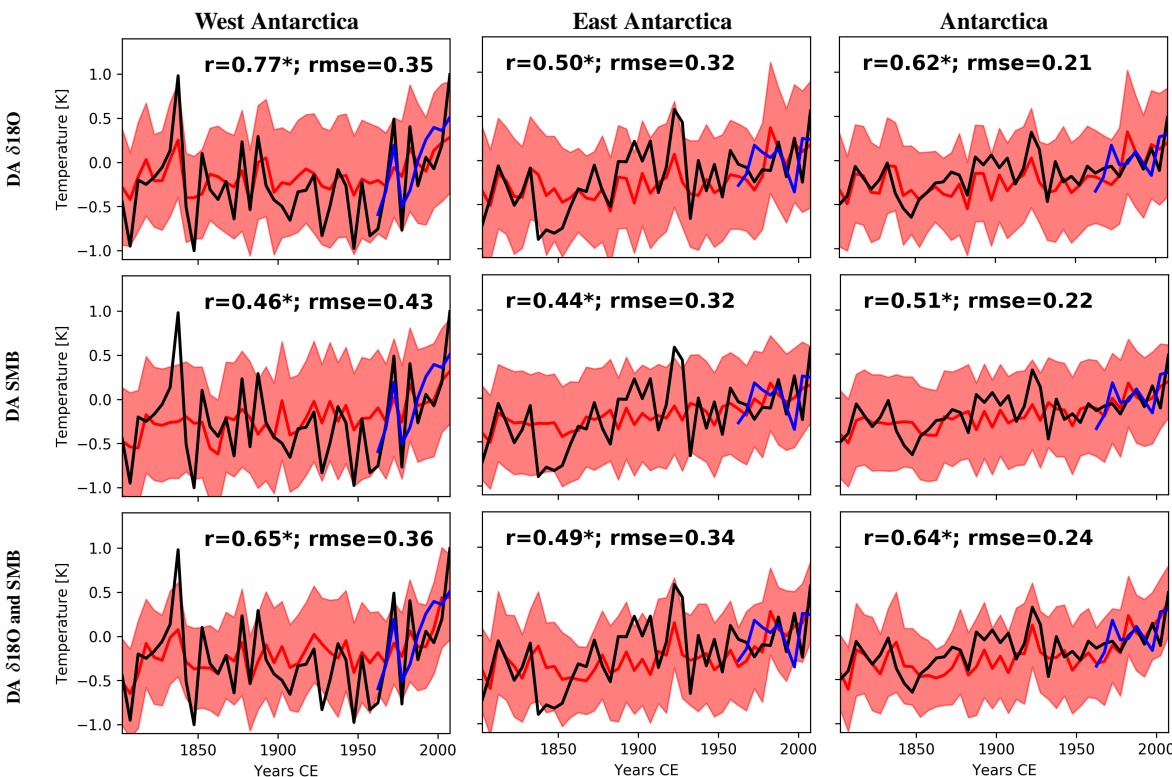

**Figure 7.** Reconstructed temperatures (5-year mean) for West Antarctica, East Antarctica and for Antarctica as a whole from data assimilation experiment (red) using the iHadCM3 outputs and $\delta^{18}O$ (Stenni et al., 2017) and SMB reconstruction (Thomas et al., 2017) as constrain in the data assimilation process. The period is 1800–2010. The surface air temperature reconstruction of Stenni et al. (2017) are represented in black and those from Nicolas and Bromwich (2014) are in blue. *DA $\delta^{18}O$* (first row) is the data assimilation experiment using only the $\delta^{18}O$ data to constrain the model while *DA SMB* (second row) uses the SMB reconstruction and *DA $\delta^{18}O$ and SMB* (third row) uses both. For each experiment and each region, the correlation (*r*) between the reconstruction based on ice cores (in black) and that based on data assimilation is computed (in red). The shaded areas represent $\pm 1$ standard deviation of the model particles. Stars represent the statistically significant correlation (p-value<0.05).

(p-value<0.1) for West Antarctica and r=0.60 (p-value<0.1) for East Antarctica. Assimilating SMB thus provides a more spatially robust temperature reconstruction than when assimilating $\delta^{18}O$. When both $\delta^{18}O$ and SMB are taken into account in the data assimilation process, the skill of the surface air temperature reconstructions for the two sub-Antarctic regions is higher (r=0.72 and 0.61 for West Antarctica and for East Antarctica respectively, both significant) than when assimilating separately the $\delta^{18}O$ or the SMB. Moreover, the only reconstruction that provides statistically significant results for all the regions (West, East and the entire Antarctica; p-value<0.1) is when both $\delta^{18}O$ and SMB are assimilated, implying that assimilating both proxies offers more robust results than only assimilating one of them.

When looking at the linearly detrended time series, our final reconstruction (i.e. when $\delta^{18}O$ and SMB are assimilated) displays a null correlation with observed surface air temperature (p-value=0.99) for West Antarctica, but the correlation remains high for East Antarctica (r=0.60; p-value=0.07). During the 1958–2012 period, a significant warming is observed in West Antarctica while no significant change is noticed for East Antarctica (Nicolas and Bromwich, 2014). Consequently, data as-
similation tends to reproduce the warming for West Antarctica and the inter-annual variability for East Antarctica, explaining our different results between the original and detrended time series. Additionally, as well as our reconstruction based on only $\delta^{18}O$, the correlation of the detrended $\delta^{18}O$-based temperature reconstruction of Stenni et al. (2017) with the observed one for East Antarctica is non-significant and negative suggesting that SMB constitutes a better proxy than $\delta^{18}O$ for surface air temperatures, at least at the inter-annual time-scale (see Tab. S2).
Regarding surface air temperature trends over the last two centuries, our reconstructions displays an increase of 0.02°C per decade for West Antarctica and 0.023°C per decade for East Antarctica, which finally leads to an increase of 0.022°C per decade for Antarctica as a whole (all statistically significant). For the 1961–2010 period, our reconstruction is able to simulate the observed contrast between West and East Antarctica (0.22 °C per decade (significant) and 0.053 °C per decade (not significant), respectively, for Nicolas and Bromwich (2014) compared with 0.1 °C per decade and 0.06 °C per decade, respectively,
for our reconstruction, both significant). The resulting contrast in our reconstruction is thus less large than observed (see Tab. S3 for details). However, because of the short time period considered, these values can highly vary depending on the time interval chosen (not shown).

### 4.3.2   SMB reconstruction

Constraining the model with the $\delta^{18}O$ data leads to a poor SMB reconstruction, especially for West Antarctica (correlation
coefficient of 0.29; Fig. 8). Moreover, the constraint derived from observed $\delta^{18}O$ on SMB is weak as illustrated by the large error band of the reconstruction (estimated by the weighted variance of the particles with non-zero weight). When assimilating both $\delta^{18}O$ and SMB, the SMB reconstruction is in good agreement with the reconstruction of Thomas et al. (2017) as expected.

**Table 2.** SMB trends over grounded (i.e. excluding ice shelves) West Antarctica, East Antarctica and Antarctica as a whole from 1) our reconstruction based on data assimilation using iHadCM3 outputs and, SMB and $\delta^{18}O$ data in the data assimilation procedure; 2) Medley and Thomas (2019); 3) RACMO2 outputs for various time intervals (in Gt year$^{-2}$). Stars stand for statistically significant trends at 5% level.

| | In this study | | | Medley and Thomas (2019) | | | RACMO2 |
|---|---|---|---|---|---|---|---|
| | 1801 – 2000 | 1957 – 2000 | 1979 – 2000 | 1801 – 2000 | 1957 – 2000 | 1979 – 2000 | 1979 – 2000 |
| West Antarctica | 0.07 | 0.82* | 1.6 | 0.1 | 1.3 | 1.7 | 2.0 |
| East Antarctica | 0.19* | -0.13 | -3.3* | 0.3* | -0.4 | -4.5* | -3.7 |
| Antarctica | 0.26* | 0.70 | -1.7 | 0.4* | 1.0 | -2.7 | -1.7 |

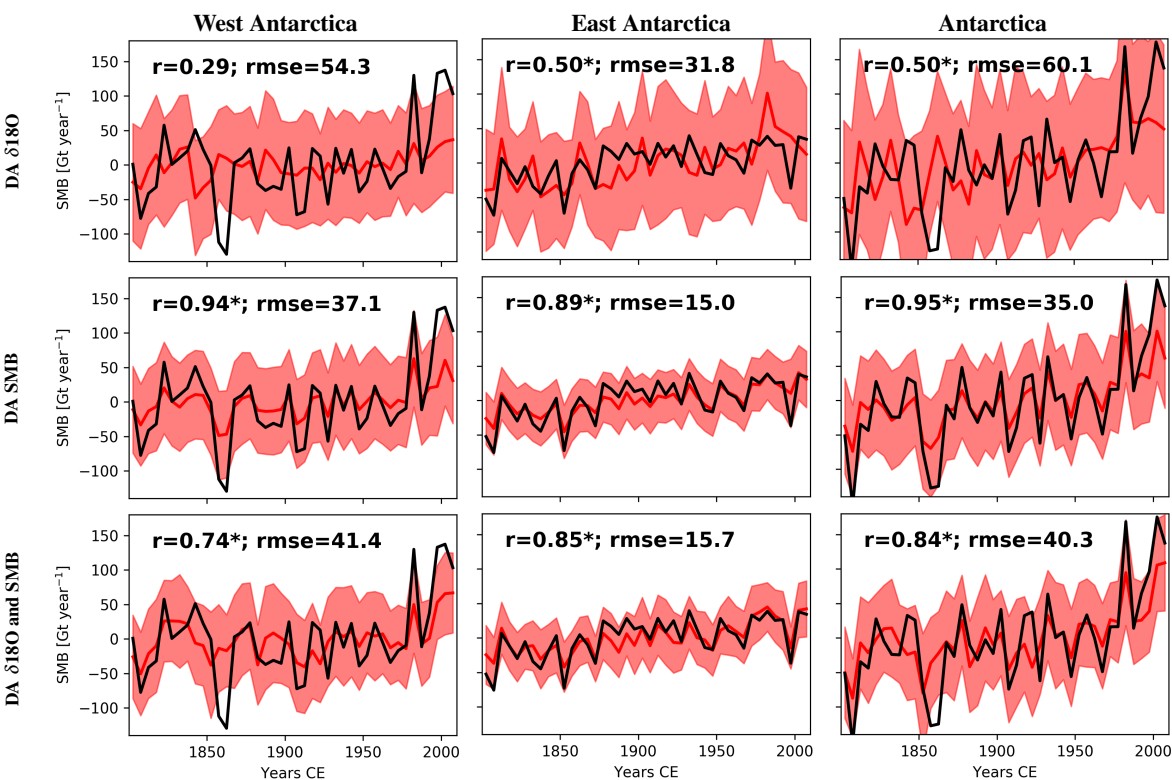

**Figure 8.** Reconstructed SMB (5-year mean) for West Antarctica, East Antarctica and Antarctica as a whole from data assimilation experiment using the iHadCM3 outputs and $\delta^{18}O$ (Stenni et al., 2017) and SMB reconstruction (in black; Thomas et al., 2017) as constrain in the data assimilation process. The period is 1800–2010. *DA $\delta^{18}O$* (first row) is the data assimilation experiment using only the $\delta^{18}O$ data to constraint the model while *DA SMB* (second row) uses the SMB reconstruction and *DA $\delta^{18}O$ and SMB* (third row) uses both. For each experiment and each region, the correlation (*r*) between the reconstruction based on ice cores (in black) and that based on data assimilation is computed (in red). The shaded areas represent $\pm$ 1 standard deviation of the model particles. Stars represent the statistically significant correlation (p-value<0.05).

According to this data assimilation-based SMB reconstruction, the grounded AIS SMB (i.e. SMB over the AIS excluding ice shelves) has increased at a 0.26 Gt year[-2] pace (p-value<0.05) during the 1801–2000 period (Tab. 2) and 0.70 Gt year[-2] (p-value>0.05) for the 1957–2000 period. Over this latter period, West Antarctica has witnessed a significant increase of 0.82 Gt year[-2] while East Antarctica displays a decrease of 0.13 Gt year[-2] (p-value>0.05). Unlike West Antarctica, the non
5   statistical significance of the SMB trend for East Antarctica might imply that internal variability currently plays a large role in the SMB variability there (e.g. Jones et al., 2016). However, if we focus on the shorter 1979–2000 period, a significant decrease is obtained for East Antarctica (-3.3 Gt year[-2]; p-value <0.05) while it is still positive for West Antarctica (1.6 Gt

year$^{-2}$; p-value>0.05), which is consistent with RACMO2 outputs (-3.7 Gt year$^{-2}$ for East Antarctica and 2.0 Gt year$^{-2}$ for West Antarctica, both not significant).

## 5 Discussion and conclusions

This paper discusses the AIS SMB over the last two centuries and its links with surface air temperature in reconstructions and model simulations. The analysis of the relationship between SMB and surface air temperature in models and in ice core reconstructions highlighted the covariance between both variables that can potentially be used to reconstruct past changes. The relevance of SMB in the reconstruction of surface air temperature in Antarctica is based on a relatively simple concept: Antarctic precipitation originates mainly from lower latitudes, in the form of warm and wet air masses (Goodwin et al., 2016; Turner et al., 2016; Clem et al., 2018). Nevertheless, $\delta^{18}$O also provides useful temperature-related information that can be used to complement the information provided by SMB, such as changes in moisture origin (e.g. Holloway et al., 2016a). Our analyses pointed out significant model-data discrepancies in the SMB-surface air temperature relationship. On the one hand, models show a strong correlation between $\delta^{18}$O and SMB for many Antarctic regions and, on the other hand, the reconstructions based on ice cores display a weak relationship. Furthermore, unlike previous studies (e.g. Frieler et al., 2015) who suggest an increase of the SMB sensitivity to surface air temperature for the future in Antarctica ($\sim$ 40%), we show that the current sensitivity is not exceptionally high compared to the last 200 years, according to CMIP5 models.

These large discrepancies between model results and reconstructions can be explained by different factors. The GCMs may have biases in the simulated temperature changes. For example, as shown by Klein et al. (2019), GCMs display on average a homogeneous warming over Antarctica during the last decades while observations mainly show warming for West Antarctica with no significant change for East Antarctica. Additionally, climate model simulations generally display a warming starting in the 19th century in Antarctica while it begins much later in proxy-based reconstructions (Abram et al., 2016). This suggests that reconstructions underestimate the response to anthropogenic forcing or that climate models overestimate it. In this latter case, this may contribute to an overestimation of the contribution of the simple thermodynamic link between temperature and precipitation and thus snow accumulation while it underestimates the role of changes in atmospheric circulation variability (Abram et al., 2016; Klein et al., 2019; PAGES 2k-PMIP3 group, 2015). Nevertheless, by removing the linear trend of time series, we obtained similar results. Models may also neglect processes such as blowing snow that can reduce the correlation between temperature and SMB. On the other hand, RACMO2, which includes a simple representation of blowing snow and is nudged to observed temperature and large-scale circulation changes, displays similar correlations to that of the GCMs. Another hypothesis is that differences could rather arise from uncertainties in the reconstructions. To understand the potential origin of the disagreements between model results and reconstructions over the last millennium, Neukom et al. (2018) used pseudoproxy experiments. They found that uncertainties in the reconstructions and the data sampling could be an explanation for many observed discrepancies between models and reconstructions.

By analyzing isotope-enabled climate models, we showed that on average over the models, the relationship between SMB and surface air temperature is often higher (or at least equivalent) and more stable than the one between surface air temperature

and $\delta^{18}$O. This is true both on the continental and regional scale. Unlike SMB, $\delta^{18}$O can be subject to large uncertainties linked to precipitation seasonality (Sime et al., 2008) or changes in moisture origins (Holloway et al., 2016a), which can explain the weaker correlations.

Our data assimilation experiments confirm the benefits of using both proxies – SMB and $\delta^{18}$O – to reconstruct surface air

temperature. When assimilating both $\delta^{18}$O and SMB data, the resulting reconstruction shows a higher correlation with observed surface air temperature over the period 1958–2010 (i.e. Nicolas and Bromwich, 2014) for the entire Antarctic continent (r=0.73) than the one obtained with the reconstruction based on the statistical method of Stenni et al. (2017; r=0.57). The difference is larger for East Antarctica, where the reconstruction skill is enhanced by incorporating SMB data (r=0.61 for our reconstruction against 0.10 for the reconstruction of Stenni et al., 2017). For West Antarctica, our reconstruction is very similar to Stenni et al.

(2017)'s statistical method. This improvement can be explained by the large uncertainties in $\delta^{18}$O data for East Antarctica, probably because of the low amount of ice cores and low snow accumulation in those areas. In comparison to Stenni et al. (2017) and Klein et al. (2019), who obtain a higher surface air temperature trend over the last two centuries for East Antarctica (0.03 °C per decade and 0.018 °C per decade respectively, both significant) than for West Antarctica (0.011 °C per decade and 0.01 °C per decade respectively, both not significant), our data assimilation-based reconstruction reveals similar surface air

temperature trends for both regions (0.02 °C per decade and 0.023 °C per decade respectively, both significant). However, over the entire continent, the trend is almost the same between the different datasets (0.022 °C per decade in this study (significant), 0.019 °C per decade for Stenni et al., 2017, significant, and 0.016 °C per decade for Klein et al., 2019, not significant). Over the last decades (1961–2010), all the reconstructions are able to reproduce the observed contrast between West Antarctica (large warming) and East Antarctica (weak warming), but overall, they underestimate it (see Tab. S3 for details). Our reconstruction

displays smaller variance in time than the reconstruction from Stenni et al. (2017), which is a standard characteristic of estimates based on data assimilation using only a few uncertain data (e.g. Goosse et al., 2010).

Regarding changes in SMB over the last two centuries, our reconstruction shows large regional differences in SMB trends, both in magnitude and in signs, in accordance with Medley and Thomas (2019; Fig. S4) who used the same ice core dataset but a different method. While they obtain a statistically significant SMB increase of 0.4 Gt year$^{-2}$ over the grounded AIS for 1801–

2000, our result suggests a weaker increase (0.26 Gt year$^{-2}$; p-value<0.001; see Tab. 2 for details). A similar underestimation is noticed for the 1957–2000 period, (1.0 Gt year$^{-2}$ for Medley and Thomas (2019), not significant, and 0.70 Gt year$^{-2}$ for our reconstruction, p-value=0.130). Over the last decades (1979–2000), both Medley and Thomas (2019) and our results reveal that grounded West Antarctica gains mass at its surface (1.6 Gt year$^{-2}$ in this study and 1.7 Gt year$^{-2}$ for Medley and Thomas, 2019, both not significant) while grounded East Antarctica has experienced a very large SMB decrease (-3.3 Gt year$^{-2}$ and -4.5

Gt year$^{-2}$ respectively, both significant), which is consistent with the value obtained in the RACMO2 outputs (2.0 Gt year$^{-2}$ for West Antarctica -3.7 Gt year$^{-2}$ for East Antarctica, both not significant).

More generally, in contrast to statistical methods, data assimilation ensures that reconstructions are compatible with the physics of the system as represented in the models chosen. Although it is not possible to independently evaluate our SMB reconstruction, our good results regarding surface air temperatures and SMB reconstructions suggest that the strong simulated

correlation between surface air temperatures and SMB in GCMs is not a model artefact. This is supported by a strong link

between these two variables in observations when using snow accumulation data from Thomas et al. (2017) and surface air temperatures from Nicolas and Bromwich (2014), in particular for East Antarctica (r=0.82, statistically significant). Therefore, our study shows that SMB records seem to be a relevant proxy in reconstructing surface air temperature in complementary with $\delta^{18}O$ records. Since only a few records are available before the instrumental period over Antarctica, any relevant record to reconstruct the Antarctic climate and more specifically surface air temperature is welcome. Additionally, data assimilation appears particularly well adapted for reconstructing surface air temperatures as the covariance between variables is obtained directly from climate models that explicitly include physical processes while statistical approaches restrict the problem to empirical linear relationships. By using data assimilation, no assumption such as stationarity or long calibration periods is required to estimate the link between variables, assumptions whose validity can strongly vary in time and space (Klein et al., 2019). However, to get a skillful data assimilation-based reconstruction, it is essential that the selected climate models have an adequate representation of climate variability and that good uncertainty estimates are available for the chosen datasets.

## Appendix A: Characteristics of GCMs

**Table A1.** PMIP3/CMIP5 GCMs characteristics and references.

| Model name | Atmospheric model resolution (lat $\times$ lon) | Number of simulations for 850–1850 period | Number of simulations for 1850–2005 period | Reference |
|---|---|---|---|---|
| BCC-CSM1-1 | 64 $\times$ 128 | 1 | 3 | Wu et al. (2014) |
| CCSM4 | 192 $\times$ 288 | 1 | 6 | Gent et al. (2011) |
| CESM1-CAM5 | 96 $\times$ 144 | 12 | 12 | Otto-Bliesner et al. (2015) |
| CSIRO-Mk3L-1-2 | 56 $\times$ 64 | 1 | 1 | Rotstayn et al. (2010) |
| GISS-E2-R | 90 $\times$ 144 | 1 | 6 | Schmidt et al. (2014) |
| HadCM3 | 73 $\times$ 96 | 1 | 10 | Turner et al. (2006) |
| IPSL-CM5A-LR | 96 $\times$ 96 | 1 | 6 | Dufresne et al. (2013) |
| MPI-ESM-P | 96 $\times$ 192 | 1 | 2 | Stevens et al. (2013) |
| MRI-CGCM3 | 160 $\times$ 320 | 1 | 3 | Yukimoto et al. (2012) |

*Code and data availability.* The resulting Antarctic SMB and surface air temperature reconstructions will be available when the manuscript is accepted. All CMIP5/PMIP3 model simulations can be directly downloaded on http://pcmdi9.llnl.gov. iHadCM3 data are available by request to Max Holloway (Max.Holloway@sams.ac.uk). ECHAM5-wiso data covering the 1871–2011 period can be downloaded from https://doi.org/10.5281/zenodo.1249604. Products from the ECHAM5/MPI-OM model simulation are available by request to Jesper Sjolte (jesper.sjolte@geol.lu.se). RACMO2 data are available by request to Jan Lenaerts (Jan.Lenaerts@Colorado.EDU). $\delta^{18}O$, surface air temperature and SMB reconstructions are stored at UK Polar Data Centre and at NOAA World Data Center for Paleoclimatology (https:

//www.ncdc.noaa.gov/paleo-search/study/22589), or by a request from Elizabeth R. Thomas (lith@bas.ac.uk). Antarctic observed surface air temperatures are available at http://polarmet.osu.edu/datasets/Antarctic_recon/.

*Competing interests.* The authors declare no competing interests.

*Acknowledgements.* We would like to thank Nathan Steiger and Jesper Sjolte for the isotopic model outputs. We would also like to thank
5   Marie Cavitte for her feedback of our study. We acknowledge the World Climate Research Programme Working Group on Coupled Modelling, which is responsible for CMIP, and we thank the climate modelling groups for producing and making available their model outputs. This work was supported by the Belgian Research Action through Interdisciplinary Networks (BRAIN-be) from Belgian Science Policy Office in the framework of the project "East Antarctic surface mass balance in the Anthropocene: observations and multiscale modelling (Mass2Ant)" (Contrat n ° BR/165/A2/Mass2Ant). Quentin Dalaiden is Research Fellow with the Fonds pour la formation á la Recherche
10  dans l'Industrie et dans l'Agronomie (FRIA-Belgium) and Hugues Goosse is the research director within the F.R.S.-FNRS.

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
