# Peer review of "How useful is snow accumulation in reconstructing surface air temperature in Antarctica? A study combining ice core records and climate models"

_The Cryosphere, 2019_

## Referee Comment (RC1) · Anonymous Referee #1 · 2 Jul 2019

The authors present the ability of CMIP5 GCMs to be used, together with ice core and d18O proxies, as a tool to reconstruct by data assimilation Antarctic temperature and SMB. They explore regionally the relation between these two variables by using different reconstruction techniques, and conclude that using both SMB and d18O proxies is most optimal. Doing this they can now better reconstruct SMB in the last two centuries.

The paper is well written, with clear figures and a new, at least to me, approach in reconstructing temperature and SMB far back in time based on physical models. The results are robust, well presented, sufficiently new and original, and I do not feel that any information is missing. I therefore strongly recommend publication in The Cryosphere.

[Figure]

However, I do have some comments on the clarity of the paper and would also recommend to make the data assimilation explanation more clear, as I will explain below.

Minor comments

P1, Title: To me the title does not really catch the main conclusions and content of the manuscript. To me, the paper comes across as a new temperature and SMB reconstruction based on a new/better technique. Do the authors feel that the main content of the paper is the link of SMB and temperature? The current title seems to "state the obvious", and did not really attract me at first to review the manuscript.

P1, Abstract, l7: This sentence is confusing, as d18O and temperature could also be the same. You mean the SMB-temperature relationship is stronger than the relationship between d18O and temperature? Maybe write out this sentence and omit the -dash.

P1, Abstract, l13: This is not clear. Which reconstruction method is used for the SMB?

P1, Abstract, general: The abstract (and title) should be reconsidered. The abstract is the first thing people read, and should be instantaneously clear. I had to re-read the abstract several times to understand it. Of course I understood it after reading the whole manuscript, but the abstract should be standalone in my opinion.

P3, l17: what is meant here with "estimated by d18O"? This relation comes out of the blue.

P10, Figure 3: Where does the very low reconstructed value for West Antarctica in ~1700 come from?

P11, Figure 4: Please change the y-axis and x-axis labels. Slope West/Slope East is unclear.

P12, Figure5: why is this shown in a contour plot? To me this is confusing. Can't you make a scatter plot (such as Fig. 7) showing the correlations?

P17, Discussion and conclusions: Same comments for this section as for the abstract:

I miss a clear emphasis on the main conclusion of the manuscript. How can these datasets be used in future work? What's the relevance of the study? What's the most important take-home message? I expect that the authors can easily strengthen the relevance of the study by giving this some extra thoughts.

---

## Referee Comment (RC2) · Anonymous Referee #2 · 20 Aug 2019

This paper from Dalaiden and co-authors addresses the question of the relationship between surface air temperature (SAT) and surface mass balance (SMB) in Antarctica, from the past 1000 years to the last decades, in view of using the SMB information for reconstructing past SAT.

Given the short and sparse observational coverage in Antarctica, reconstruction of the Antarctic climate further than the last decades rely on the interpretation of proxies. The isotopic composition of the snow (in particular  $\delta$ 180 in ice cores) is the most widely used proxy of SAT in Antarctica

First the authors show that the strong link between SMB and SAT, already acknowledged in the literature (e.g. Frieler et al 2015), remain valid in GCMs during the past 1000 years and the past 200 years. They also show that the relationship does not stand when considering the last two reconstructions of surface air temperature (based on ice cores  $\delta$ 180, Stenni et al., 2017) and surface mass balance (based on ice cores accumulation, Thomas et al., 2017), but does exist when using an Antarctic SAT reconstruction based on weather stations (Nicolas and Bromwich 2014, NB14) instead of the SAT reconstruction based on ice cores  $\delta$ 180.

Then the authors use isotope-enabled global climate models to perform an offline data assimilation of  $\delta$ 18O and SMB over the past 200 years. They obtain more consistent results with NB14 SAT over West and East Antarctic ice sheets when combining the assimilation of  $\delta$ 18O and SMB.

I think using both SMB and  $\delta$ 18O for reconstructing SAT with an assimilation method is novel and relevant for the cryosphere and climate community. The overall presentation is clear and figures are nicely shaped. Conclusions seem robust and interesting. However I have some concerns about some of the interpretations, and I also have comments on the methodology. Therefore I recommend this article to be published after addressing the following issues.

**Major**

1) I think the GCM evaluation is of interest, in particular the plots comparing SMB by elevation bins, but I disagree with the conclusion that GCM are doing a good job in Antarctica. I think this is not a critical point for this study, so the authors should minimize or remove the section about GCM evaluation (Section 4.1, one or two sentences and citing supplementary would be enough) and extend the analysis on the SMB/SAT relationship (Section 4.2). Fig. 2 is not necessary, Fig. 3 and Fig. 4 could be moved to the SAM/SAT section, Fig. 4 could be extended with a scatterplot comparing SMB/SAT sensitivity factors (% K-1) of West vs East. This way the result section would follow the plan detailed in the introduction: i) SMB/SAT in GCMS over the past millennia and
centuries ii) data assimilation for the past centuries.

In detail:

\* Abstract "Here, we show that Global Climate Models (GCMs) can reproduce the present-day (1979–2005) AIS SMB and the temporal variations over the last two centuries."

\* P17 "The GCMs are able to simulate relativity well the current AIS SMB" -> Should be rephrased or removed (see hereafter).

\* P8 "Overall, the AIS SMB simulated by GCMs is in good agreement with the SMB simulated by the regional climate model RACMO2 over the last decades (1979–2005, R2 = 0.53; Fig. 2 and S1 for the SMB of each model)."

-> I see huge differences, spatially and integrated over the ice sheet (Fig. S1 and S2). How is computed this correlation coefficient? What is the bias?

\* P8 "Both display high values of SMB along the coast (>300 mm w.e. year-1) – especially for West Antarctica and the Antarctic Peninsula – and lower values at high elevations (e.g. the Plateau: <100 mm w.e. year-1)."

-> This is really the minimum feature a model can do, because of the general circulation and the ice sheet topography.

2) I found interpretations in contradiction with the figures.

\* P9 "Nevertheless, when analyzing the individual simulations of the ensemble performed with CESM1-CAM5, the contrast between East Antarctica and West Antarctica is as large as in recent observations (Fig. 4). This indicates that 1) the observed SMB trends between the two regions are within the range of the simulated values; 2) internal variability has an important role in the current Antarctic SMB changes."

-> Reconstruction is a clear outlier of the GCM's scatterplots, so reformulate the conclusion in agreement with your figure.

\* P12 "For most regions, the link between surface temperature and SMB (r=0.70 on
average over the seven subregions for the 1850–2000 period) is higher than that between surface temperatures and  $\delta$ 18O (r=0.55 on average over the seven subregions for the 1850–2000 period)." (...) "The results with the outputs of ECHAM5-wiso and ECHAM5/MPI-OM are similar (Figs. S6 and S7)."

-> It does not appear to be true when looking at Fig. S6 and S7: blue dots (SAT/ $\delta$ 18O) are often higher than green dots (SMB/SAT). I regret this over-interpretation and the fact that the authors focused on the iHadCM3 in the main text without specifying it and explaining this choice.

P18 "On the one hand, models show a strong correlation between  $\delta 180$  and SMB for all the Antarctic regions"

-> It's not true: red dots in Fig 7, S6 and S7. Is there a typo here? But even SAT-SMB relationship is not strong for all regions (Fig S5).

"we showed that the relationship between SMB and surface temperature is often higher than the one between surface temperature and  $\delta$ 18O. This is true both on the continental and regional scale."

-> That's not true when considering ECHAMwiso and ECHAM/MPI-OM

3) Methodology

Data assimilation (DA) must be evaluated with independent datasets. It is the case for SAT (NB14 is not assimilated) but not for SMB. The authors assimilate SMB from Thomas et al. (2017) and evaluate their results with Thomas et al. (2017). I suggest to use independent and annually resolved datasets, such as the radar transects resolved annually in West Antarctica (Medley et al. 2014 https://doi.org/10.5194/tc-8-1375-2014) and stake line transects (JARE, CHINARE).

\* P19 "Considering our good results regarding surface temperatures and SMB reconstructions,"

-> This sentence is not fair if you evaluate your result with the data you assimilate.

\* P19 "our data assimilation-based reconstructions suggest that the strong simulated
correlation between surface temperatures and SMB in GCMs is not a model artefact" -> DA is a weighted average, so if the SMB-SAT relationship exists in the models, isn't it conserved in the reconstruction by construction?

**4) A remark**

Results of data assimilation seem less variable than the other reconstructions (Fig 8 and Fig 9). Is it due to the assimilation method? What is the confidence on the DA temporal variability?

**Minor**

**Abstract**

"with a linear correlation coefficient with the observed surface temperatures (1958– 2010 CE) of 0.73"

I don't think this number is meaningful, I suggest to remove it.

**P2**

"(Rignot et al., 2011)"

Update with Rignot et al. (2019) https://www.pnas.org/content/116/4/1095

"(Wouters et al., 2013)." Idem, update the reference.

"from stable isotope ratios of oxygen" From water stable isotopes, and in particular  $\delta$ 18O

**Р3**

"According to Monaghan et al. (2008), the observed sensitivity of Antarctic snowfall accumulation to surface temperature was about 5% K-1 during the 1960–1999 period." Why Monaghan and not a most recent and complete reference? (e.g. Frieler 2015)

"These results suggest that in some regions, especially along the AIS coasts, the vari-
ability of thermodynamic processes (such as the Clausius-Clapeyron effect) on SMB is dominated by the large-scale atmospheric circulation, limiting the correlation with  $\delta 180."$

Do you mean: SMB variability is dominated by large-scale atmospheric circulation rather than by thermodynamic processes?

"While the statistical methods classically used to infer past surface temperature (see for instance Stenni et al., 2017) rely on the length of the calibration period, on the quality of the record during this period, and on the stationarity of the link between the proxy and the variable of interest, which can be strong assumptions in the case of the  $\delta$ 18O-temperature relationship (Klein et al., 2019), data assimilation does not." Doesn't data assimilation rely on the quality of the assimilated record too? One step further, a short sentence about the limits of the assimilation method is missing, to be fair. E.g. changes in the number and quality of assimilated data?

**Ρ4**

"The simulation of ECHAM5-wiso, which only includes an atmospheric component, was performed by Steiger et al. (2017) and covers the period 1871–2011 CE at âLij 1° resolution. The model is driven by the sea surface temperature and sea ice from the Rayner et al. (2003) dataset."

You have to mention that the Rayner et al. (2003) dataset is not relevant before 1973: "2.1.3. Antarctic Atlas Climatologies Before the advent of satellite-based imagery in 1973, sea ice concentration data for the Antarctic are not available, and sea ice extent data are not readily available for individual months, seasons or years, although some visible and infrared data do exist for 1966–1972 [Zwally et al., 1983] and some undigitized charts reside in national archives (e.g., V. Smolyanitsky, personal communication, 2002). Readily available information was limited to two historical climatologies of sea ice extent. Therefore our sea ice concentration analysis before 1973 is derived indirectly, and does not include any interannual variability, though there are some trends resulting from the differences between climatologies for different periods."
"Comparisons of the results of these three isotope-enabled models with modern  $\delta$ 180 observations indicate that they all reproduce the main characteristics of the spatial distribution of the isotopic composition of precipitation over Antarctica (see reference for each model)."

Add a word about their known biases.

**Ρ5**

"(4) the output of RACMO2 for the AIS SMB agrees very well with available measurements (correlation coefficient with observations of 0.9; van Wessem et al., 2018)." A high correlation coefficient alone is not a proof of good performance. Correlation can be equal to one with a very large bias.

**P6**

"This temporal averaging reduces uncertainties in dating linked to the noise induced by non-climatic processes (e.g. Laepple et al., 2018; Fan et al., 2014)." The temporal averaging is not described before, and I understood latter in the paragraph that you were talking about the 5-year and 10-year average. The whole paragraph is strangely shaped, please rephrase.

**Ρ7**

"each ensemble member, called particle, is compared to the proxy-based reconstruction by computing its likelihood, taking into account data uncertainties." Give a description of this likelihood function. How do you compute it?

**P8**

" The median of the SMB over the entire AIS simulated by CMIP5 models is 1.16A median computed from 12 values is not robust. This number is hiding large discrepancies between the models.

\*Figure 2:\* You show the average while above you give the number for the median.
"who have shown that due to the lower spatial resolution of GCMs in comparison to the regional model, SMB is underestimated at the coasts while an overestimation occurs in the interior of the ice sheet."

Resolution might play a role but model's physics also plays a major role. E.g. Fig S1 shows that MRI-CGCM3 and ECHAM-wiso have much large SMB at the margins than RACMO2, whereas they have a lower resolution.

\*Fig. S3:\* Add the isotope-enabled models

"confirming that the spatial resolution has a crucial impact on the simulated SMB." This is not convincing and not the dominant factor in my point of view.

P11 "According to these reconstructions, this sensitivity has increased a lot for the recent period (1950–2005; 15.52 Do you think it is realistic? I don't find such an increase in sensitivity in Frieler et al. (2015)?

\*Figure 6:\* I don't understand why for WAIS and AP, 'reconstructions' (black line) is lower than model mean, while for the combination of both (West Antarctica), 'reconstructions' is larger than the model mean? + typos in the legend.

**P12**

"The analysis of isotope-enabled models results reinforces this hypothesis (Fig. 7): the iHadCM3 outputs show high correlations between these two variables." In the sub-section 4.3, you only focus on the iHadCM3 outputs without explicitly announcing it and explaining why you did this choice.

**P16**

"(estimated by the weighted variance of the particles with non-zero weight)" Define this weight/metric in the method section. What is the threshold?

"When assimilating both  $\delta 180$  and SMB, the SMB reconstruction is in good agreement with the reconstruction of Thomas et al. (2017)."

TCD
As expected as Thomas is assimilated.

P18

"who suggest an increase of the SMB sensitivity to surface temperature for the future in Antarctica,"

Can you give a number?

"The GCMs may have biases in the simulated temperature changes or in their response to anthropogenic forcing."

This is very general, what are the known biases in GCMs?

"This may contribute to an overestimation of the contribution of the simple thermodynamic link between temperature and precipitation and thus snow accumulation while it underestimates the role of changes in atmospheric circulation variability." Any reference on this point?

"According to Neukom et al. (2018), uncertainties in the reconstructions (the noise in proxy data and the deficiencies in the reconstruction methods) and the data sampling could be an explanation of the observed discrepancy between models and reconstructions."

Give some key details on how it is proven.

"surface temperature over the period 1958–2010" Add the reference (Nicolas and Bromwich, 2014)

P19

"Regarding changes in SMB over the last two centuries, our reconstruction shows large regional differences in SMB trends, both in magnitude and in sign, in accordance with Medley and Thomas (2019; Fig. S12)."

A word on the fact that DA assimilate Thomas 2017, which use the same ice core dataset as in Medley and Thomas 2019? So it is not surprising that patterns are similar?
"This is supported by a strong link between these two variables in observations, in particular for East Antarctica (r=0.82, statistically significant)." Specify that is is between Thomas et al 2017 and NB14, and does not work with Stenni 2017

"By using data assimilation, no assumption such as stationarity or long calibration periods is required to estimate th link between variables" Please also include the limitations of the data assimilation method. th> the

---

## Author Comment (AC1) · 18 Sep 2019

Dear referee,

Thank you for your comments and suggestions.

We have attached our detailed responses.

Kind regards,

On behalf of all authors,

Dalaiden Quentin

Please also note the supplement to this comment:
https://www.the-cryosphere-discuss.net/tc-2019-111/tc-2019-111-AC1-supplement.pdf

---

## Author Response (AR1)

Dear authors

Thanks for submitting your responses. I would like to invite you to submit the revised manuscript for the full consideration.

Kenny Matsuoka

TC/TCD Editor

Dear Editor,

We would like to thank you for editing our manuscript. We have updated it according to the comments of the reviewers as suggested in our responses. You can find below the point by point response to each reviewer's comment as well as the final version of our manuscript with track changes.

On behalf of all co-authors,

Dalaiden Quentin

The Referee's comments below are in italics, our answer in plain font in blue

*The authors present the ability of CMIP5 GCMs to be used, together with ice core and d18O proxies, as a tool to reconstruct by data assimilation Antarctic temperature and SMB. They explore regionally the relation between these two variables by using different reconstruction techniques, and conclude that using both SMB and d18O proxies is most optimal. Doing this they can now better reconstruct SMB in the last two centuries. The paper is well written, with clear figures and a new, at least to me, approach in reconstructing temperature and SMB far back in time based on physical models. The results are robust, well presented, sufficiently new and original, and I do not feel that any information is missing. I therefore strongly recommend publication in The Cryosphere. However, I do have some comments on the clarity of the paper and would also recommend to make the data assimilation explanation more clear, as I will explain below.*

We would like to thank the Referee for the positive evaluation and for the useful comments.

*P1, Title: To me the title does not really catch the main conclusions and content of the manuscript. To me, the paper comes across as a new temperature and SMB reconstruction based on a new/better technique. Do the authors feel that the main content of the paper is the link of SMB and temperature? The current title seems to"state the obvious", and did not really attract me at first to review the manuscript.*

We agree with the referee that the title does not totally correspond to the main content of the manuscript. We have decided to change it to: "How useful is snow accumulation in reconstructing surface temperature in Antarctica? A study combining ice core records and climate models."

*P1, Abstract, l7: This sentence is confusing, as d18O and temperature could also be the same. You mean the SMB-temperature relationship is stronger than the relationship between d180 and temperature? Maybe write out this sentence and omit the -dash.*

We have changed this sentence. "We find that, on the regional scale, the modeled relationship between surface temperature and SMB is generally stronger than between temperature and $\delta^{18}O$."

*P1, Abstract, l13: This is not clear. Which reconstruction method is used for the SMB?*

We agree that this sentence is ambiguous. We have changed it by:

Finally, we provide a spatial SMB reconstruction of the AIS over the last two centuries showing 1) large variability in SMB trends at regional scale; and 2) a large SMB increase (0.82 Gt year$^{-2}$) in West Antarctica over 1957–2000 while at the same time, East Antarctica has experienced a large SMB decrease (-3.3 Gt year$^{-2}$), which is consistent with a recent reconstruction.

by:

Finally, using the same data assimilation method as for the surface temperature reconstruction, we provide a spatial SMB reconstruction for the AIS over the last two centuries showing large variability in SMB trends at regional scale, with an increase (0.82 Gt

year$^{-2}$) in West Antarctica over 1957–2000 and a decrease in East Antarctica during the same period (-3.3 Gt year$^{-2}$). As expected, this is consistent with the recent reconstruction used as a constraint in the data assimilation.

*P1, Abstract, general: The abstract (and title) should be reconsidered. The abstract is the first thing people read, and should be instantaneously clear. I had to re-read the abstract several times to understand it. Of course I understood it after reading the whole manuscript, but the abstract should be standalone in my opinion.*

As suggested by the reviewer, we have rewritten the abstract to highlight our main conclusions.

*P3, l17: what is meant here with "estimated by d18O"? This relation comes out of the blue.*

We wanted to point out here that the $\delta^{18}O$ is used as a proxy of surface temperature in some studies analyzing the link between surface temperature and SMB. Therefore, those studies (e.g. Fudge et al., 2016; Altnau et al., 2015; Philippe et al., 2016; Goursaud et al., 2019) have analyzed the link between $\delta^{18}O$ and SMB rather than the link between surface temperature and SMB. In other words, they are not based on observed surface temperature but on estimated surface temperature derived from $\delta^{18}O$.

We have changed this sentence to illustrate this:

However, some studies (Fudge et al., 2016; Altnau et al., 2015; Philippe et al., 2016; Goursaud et al., 2019) indicate that this SMB-surface temperature relationship (estimated by $\delta^{18}O$) is not always positive, and varies spatially and temporally.

by:

However, some studies using surface temperature reconstructions based on $\delta^{18}O$ data (Fudge et al., 2016; Altnau et al., 2015; Philippe et al., 2016; Goursaud et al., 2019) suggest that this SMB-surface temperature relationship is not always positive and varies spatially and temporally.

*P10, Figure 3: Where does the very low reconstructed value for West Antarctica in∼1700 come from?*

This very low value is likely related to the low number of ice cores used for the SMB composite of the West Antarctica region at this time. As shown by Thomas et al. (2017), the regional SMB composites before 1800 are based on very few records, which can lead to large uncertainties. We have decided to only display the 1800-2010 period for the reconstruction to avoid those uncertain values.

*P11, Figure 4: Please change the y-axis and x-axis labels. Slope West/Slope East is unclear.*

We agree that this plot is unclear. We have changed the plot to make it clearer (see the response to the second review).

*P12, Figure5: why is this shown in a contour plot? To me this is confusing. Can't you make a scatter plot (such as Fig. 7) showing the correlations?*

We think it is important to display the correlations between SMB and surface temperatures on a map instead of a scatter plot to keep the spatial dimension. For example, by analyzing the results for RACMO2, we observe that the coastal regions of East Antarctica display weak correlations between the two variables. Replacing this map by a scatter plot will remove this spatial information, which is important in our interpretation.

*P17, Discussion and conclusions: Same comments for this section as for the abstract:*
*I miss a clear emphasis on the main conclusion of the manuscript. How can these datasets be used in future work? What's the relevance of the study? What's the most important take-home message? I expect that the authors can easily strengthen the relevance of the study by giving this some extra thoughts.*

We will change our conclusion in order to strengthen our main findings as asked by the referee.

The Referee's comments below are in italics, our answer in plain font in blue

*This paper from Dalaiden and co-authors addresses the question of the relationship between surface air temperature (SAT) and surface mass balance (SMB) in Antarctica, from the past 1000 years to the last decades, in view of using the SMB information for reconstructing past SAT. Given the short and sparse observational coverage in Antarctica, reconstruction of the Antarctic climate further than the last decades rely on the interpretation of proxies. The isotopic composition of the snow (in particularδ18O in ice cores) is the most widely used proxy of SAT in Antarctica. First the authors show that the strong link between SMB and SAT, already acknowledged in the literature (e.g. Frieler et al 2015), remain valid in GCMs during the past1000 years and the past 200 years. They also show that the relationship does not stand when considering the last two reconstructions of surface air temperature (based on ice cores δ18O, Stenni et al., 2017) and surface mass balance (based on ice cores accumulation, Thomas et al., 2017), but does exist when using an Antarctic SAT reconstruction based on weather stations (Nicolas and Bromwich 2014, NB14) instead of the SAT reconstruction based on ice cores δ18O. Then the authors use isotope-enabled global climate models to perform an offline data assimilation of δ18O and SMB over the past 200 years. They obtain more consistent results with NB14 SAT over West and East Antarctic ice sheets when combining the assimilation of δ18O and SMB. I think using both SMB and δ18O for reconstructing SAT with an assimilation method is novel and relevant for the cryosphere and climate community. The overall presentation is clear and figures are nicely shaped. Conclusions seem robust and interesting. However I have some concerns about some of the interpretations, and I also have comments on the methodology. Therefore I recommend this article to be published after addressing the following issues.*

We would like to thank the Referee for the careful evaluation and for all the suggestions that helped to improve the manuscript.

*Major*

*1) I think the GCM evaluation is of interest, in particular the plots comparing SMB by elevation bins, but I disagree with the conclusion that GCM are doing a good job in Antarctica. I think this is not a critical point for this study, so the authors should minimize or remove the section about GCM evaluation (Section 4.1, one or two sentences and citing supplementary would be enough) and extend the analysis on the SMB/SAT relationship (Section 4.2). Fig. 2 is not necessary, Fig. 3 and Fig. 4 could be moved to the SAM/SAT section, Fig. 4 could be extended with a scatterplot comparing SMB/SAT sensitivity factors (% K-1) of West vs East. This way the result section would follow the plan detailed in the introduction: i) SMB/SAT in GCMS over the past millennia and* centuries ii) data assimilation for the past centuries.

As suggested by the reviewer, we have trimmed the GCM evaluation. The evaluation over the recent past (1979-2005; i.e. the comparison to RACMO outputs) has been moved to Supplementary Materials. However, we have kept the section on the comparison between the simulated and reconstructed (i.e. Thomas et al., 2017) SMB changes during the last two centuries. Therefore, we have adapted the title section: "Reconstructed and simulated SMB changes over the last centuries".

The Fig. 4 has been extended with a scatter plot comparing the SMB/SAT sensitivity factors:

[Figure]

[Figure]

**Figure 4.** (left) Comparison between the reconstructed and the simulated SMB trends (mm w.e./100y$^{-2}$) over the period 1950–2000 CE in West Antarctica (y axis) and East Antarctica (x axis). (right) As on the left but for SMB/SAT sensitivity factors (% K$^{-1}$). For the reconstruction, data from Thomas et al. (2017) and Nicolas and Bromwich (2014) are used.

*In detail:*
*\* Abstract "Here, we show that Global Climate Models (GCMs) can reproduce the present-day (1979–2005) AIS SMB and the temporal variations over the last two centuries."*

We have removed this sentence to stay focused in the abstract on the SMB-SAT relationship and on our reconstructions.

*\* P17 "The GCMs are able to simulate relativity well the current AIS SMB"*
*-> Should be rephrased or removed (see hereafter).*

We have removed the SMB evaluation in the discussion/conclusions section.

*\* P8 "Overall, the AIS SMB simulated by GCMs is in good agreement with the SMB simulated by the regional climate model RACMO2 over the last decades (1979–2005,R2 = 0.53; Fig. 2 and S1 for the SMB of each model)."*
*-> I see huge differences, spatially and integrated over the ice sheet (Fig. S1 and S2). How is computed this correlation coefficient? What is the bias?*

We have made a correlation plot (new figure: see below, Fig. S2) of the SMB climatology as simulated by the average of the GCMs as a function of the climatology of RACMO over the 1979-2005 period. The correlation is computed between the model mean spatial distribution (averaged over 1979-2005) and the spatial distribution of RACMO over the same period. The model mean has been interpolated on the RACMO grid to compute the correlations. The bias is the average of the difference between the GCM mean and RACMO (in mm w.e. year$^{-1}$).

[Figure]

**Figure S2.** Correlation plot of SMB climatology from GCM mean (average over all the GCMs including isotope-enabled models) as a function of SMB RACMO over the 1979–2005 period at the same location. $R^2$ is the determination coefficient and the estimation of the bias is the average of the difference between GCM mean and RACMO (in mm w.e. year$^{-1}$). Red (blue) dots are for places where the altitude is lower (higher) than 1500m. See Fig. S4 for the equivalent for each model.

Because we have added the isotope-enable models in the evaluation, we have updated the following sentence:

"The mean of the SMB over the entire AIS simulated by the selected CMIP5 models is 87 Gt year -1 higher than the SMB simulated by RACMO2 (relative bias: -3.7%; see Fig. S2 for the integrated SMB over the entire AIS for each model)."

by:

"The mean of the SMB over the entire AIS simulated by the selected models (including isotope-enable models) is 6.4 mm w.e. year -1 lower than the SMB simulated by RACMO2 over the 1979-2005 period (relative bias: -3.4%; see Fig. S4 for the correlation plots for each model and Fig. S5 for the integrated SMB over the entire AIS for each model)."

*\* P8 "Both display high values of SMB along the coast (>300 mm w.e. year-1) – especially for West Antarctica and the Antarctic Peninsula – and lower values at high elevations (e.g. the Plateau: <100 mm w.e. year-1)."*
*-> This is really the minimum feature a model can do, because of the general circulation and the ice sheet topography.*

Yes, we totally agree with your remark, but we think that it is important to notice the main Antarctic SMB pattern. Therefore, we have added "As expected" at the beginning of the sentence to show that is not something surprising.

*2) I found interpretations in contradiction with the figures.*
*\* P9 "Nevertheless, when analyzing the individual simulations of the ensemble performed with CESM1-CAM5, the contrast between East Antarctica and West Antarctica is as large as in recent observations (Fig. 4). This indicates that 1) the observed SMB trends between the two regions are within the range of the simulated values; 2) internal variability has an important role in the current Antarctic SMB changes."*
*-> Reconstruction is a clear outlier of the GCM's scatterplots, so reformulate the conclusion in agreement with your figure.*

We have changed the paragraph following the suggestion to be in better agreement with the figure:

"When analyzing the ensemble of simulations performed with CESM1-CAM5, the ensemble mean also shows a relatively homogeneous increase, but some simulations display a contrast between East Antarctica and West Antarctica close to the one observed in the reconstruction (Fig. 3). This suggests that internal variability has a dominant contribution in the current Antarctic SMB changes and might explain why the observed contrast between East and West Antarctica is only present in a few simulations."

*P12 "For most regions, the link between surface temperature and SMB (r=0.70 on average over the seven subregions for the 1850–2000 period) is higher than that between surface temperatures andδ18O (r=0.55 on average over the seven subregions for the 1850–2000 period)." (...) "The results with the outputs of ECHAM5-wiso and ECHAM5/MPI-OM are similar (Figs. S6 and S7)."*
*-> It does not appear to be true when looking at Fig. S6 and S7: blue dots (SAT/δ18O) are often higher than green dots (SMB/SAT). I regret this over-interpretation and the fact that the authors focused on the iHadCM3 in the main text without specifying it and explaining this choice.*

We mostly focused on iHadCM3 outputs and not on the other isotope-enable models in the main the text because, in contrast to the other isotope-enabled models (ECHAM5-wiso and ECHAM5/MPI-OM), iHadCM3 offers an ensemble of simulations which is a significant advantage for data assimilation. Indeed, dealing with an ensemble of simulations allows increasing the probability to find a good match between the assimilated records and model results during the assimilation process.

Regarding the ECHAM5-wiso and ECHAM5/MPI-OM models, we have modified the figures S6 and S7 to replace them by the Figure S9:

[Figure]

**Figure S9.** 5-year mean correlations between surface temperature and $\delta^{18}O$ (blue) and, SMB and surface temperature (green) for the seven Antarctic regions for the entire period simulation (1871–2010 for ECHAM5-wiso and 801–2000 for ECHAM5/MPI-OM).

This new figure allows for an easier comparison between the potential of SMB and $\delta^{18}O$ in reconstructing regional surface temperatures. As the reviewer mentioned, the results of ECHAM models are a little different than those of iHadCM3. We thus propose to discuss in more details those results of the ECHAM in the main text:

"The results of ECHAM5-wiso and ECHAM5/MPI-OM confirm this strong link between SMB and temperature but, in contrast to iHadCM3, the correlations are not systematically higher than between $\delta^{18}$O and temperature (Fig. S9). When analyzing the long ECHAM5/MPI-OM simulation (800–2000), the relationship between SMB and surface temperature is generally higher than between $\delta^{18}$O and surface temperature but the difference is small. For some regions, the SMB-surface temperature link is much higher than the $\delta^{18}$O-surface temperature link but it is weaker for other regions. In contrast to the $\delta^{18}$O-surface temperature link, the SMB-surface temperature is less spatially variable (minimum regional correlation is 0.54 against 0.07 for the $\delta^{18}$O-surface temperature link)."

*P18 "On the one hand, models show a strong correlation between δ18O and SMB for all the Antarctic regions"-*
*> It's not true: red dots in Fig 7, S6 and S7. Is there a typo here? But even SAT-SMB relationship is not strong for all regions (Fig S5).*

Indeed, we made a mistake here (it is the SAT-SMB relationship and not the $\delta^{18}$O-SMB relationship that shows a strong correlation for all regions). Thank you for that.

We propose to replace "for all the Antarctic regions" by "many Antarctic regions".

*"we showed that the relationship between SMB and surface temperature is often higher than the one between surface temperature andδ18O. This is true both on the continental and regional scale."*
*-> That's not true when considering ECHAMwiso and ECHAM/MPI-OM*

Even though the ECHAM models do not always display stronger regional correlations between SMB and surface temperature than between $\delta^{18}$O and surface temperature, on average over all the isotope-enable models, the SMB-surface temperature link is stronger (90% of the time for iHadCM3, 80% for ECHAM/MPI-OM and 50% for ECHAM5-wiso) and more stable than the $\delta^{18}$O -surface temperature link. We propose to modify slightly this sentence:

"By analyzing isotope-enabled climate models, we show that the relationship between SMB and surface temperature is often higher than the one between surface temperature and $\delta$ 18 O."

by:

"By analyzing isotope-enabled climate models, we showed that on average over the models, the relationship between SMB and surface temperature is often higher (or at least equivalent) and more stable than the one between surface temperature and $\delta^{18}$O."

*3) Methodology*
*Data assimilation (DA) must be evaluated with independent datasets. It is the case for SAT (NB14 is not assimilated) but not for SMB. The authors assimilate SMB from Thomas et al. (2017) and evaluate their results with Thomas et al. (2017). I suggest to use independent and annually resolved datasets, such as the radar transects resolved annually in West Antarctica (Medley et al. 2014 https://doi.org/10.5194/tc-8-1375-2014) and stake line transects (JARE, CHINARE).*
*\* P19 "Considering our good results regarding surface temperatures and SMB reconstructions,"*
*-> This sentence is not fair if you evaluate your result with the data you assimilate.*

We totally agree with the reviewer. Our goal is to propose a new reconstruction method for surface temperature. It is thus needed to evaluate this new reconstruction with an independent dataset. Unfortunately, we did not find any suitable dataset to evaluate our data assimilation-based

reconstruction. The radar transects that you suggest (Medley et al., 2014) cover a small part of the West Antarctic Ice Sheet over the 1985-2009 period. It is thus not possible to make an evaluation at the scale of Antarctica. Furthermore, because we applied a 5-year smoothing on our SMB and surface temperature reconstruction to remove the non-climatic noise, any validation would be based on a too small sample (applying a 5-year smoothing on the NB2014 dataset which covers the 1958-2012 period reduces the time series to 12 points which is already low for making correlations).

This absence of independent datasets forbids us to evaluate the skill of the new reconstruction. The comparison of our data assimilating-based SMB reconstruction to Thomas et al. (2017) is thus only done to check if the reconstruction is consistent with all the input information or if major incompatibilities are present. If model results (used as prior) and data are too different or if the uncertainty is not well estimated, the particle filter may degenerate. The resulting reconstruction can also be far away from the assimilated records if there is no model result that fits with the signal recorded in those data. Our comparison to Thomas et al. (2017) is not independent but at least shows that our reconstruction is consistent with Thomas et al. (2017). This is indeed expected but good to verify.

We specified in the experimental design (section 3.2) that we are not able to independently evaluate our SMB reconstruction:

"SMB estimates are also available for the last decades (e.g. Medley et al. 2014), but they cover a too short period or have a too small spatial coverage to provide an independent validation of our reconstruction. It is thus not possible to estimate if the assimilation of SMB and $\delta^{18}O$ measurements provides an improvement for this field."

We have also specified in the discussion/conclusions section that we cannot independently simulate our SMB reconstruction:

"Although it is not possible to independently evaluate our SMB reconstruction, our good results regarding surface temperatures and SMB reconstructions suggest that the strong simulated correlation between surface temperatures and SMB in GCMs is not a model artefact."

*\* P19 "our data assimilation-based reconstructions suggest that the strong simulated correlation between surface temperatures and SMB in GCMs is not a model artefact"
-> DA is a weighted average, so if the SMB-SAT relationship exists in the models, isn'tit conserved in the reconstruction by construction?*

Yes, this link should be preserved as the reconstruction is based on the covariance between those two variables as displayed in models. However, if the models were overestimating this link, the particle filter would give more weight to the model results that display the weakest correlation. Furthermore, the increased skill of the surface temperature reconstruction when including SMB data also indicates that the model covariance is bringing additional information. This is not a formal proof. This is the reason why in the corresponding sentence, we propose to use 'suggest' (see the new proposed sentence just above), but it remains consistent with the fact that the strong correlation between SMB and surface temperature is not a model artefact.

*4) A remark
Results of data assimilation seem less variable than the other reconstructions (Fig 8 and Fig 9). Is it due to the assimilation method? What is the confidence on the DA temporal variability?*

The mean reconstruction provided by data assimilation may underestimate the variability if the data is too uncertain or if there is not enough data. In the extreme case when you have no data (or with

data displaying a very large uncertainty), the particle filter will just give a reconstruction that is the model ensemble mean which consists here, because of the experiment design, in a value of zero for the whole period. However, in that case, the uncertainty of the ensemble would be very large, and this of course must be taken into account when discussing the temporal variability of the reconstruction. More specifically, with only a few uncertain data, it is expected that the reconstruction based on our data assimilation method may show less variance than reconstructions provided by some other methods (as observed previously; e.g. Goosse et al. 2010). Nevertheless, we did not discuss much this point in the manuscript as it critically depends on the uncertainty of the input data, that is itself not well known.

*P8*
*"The median of the SMB over the entire AIS simulated by CMIP5 models is 1.16"*
*A median computed from 12 values is not robust. This number is hiding large discrepancies between the models.*

We have replaced the median by the mean in the text (absolute and relative biases):

"The SMB integrated over the entire AIS is 87 Gt year $^{-1}$ higher for the mean of the selected CMIP5 models than in RACMO2 (relative bias: -3.7%; see Fig. S2 for the integrated SMB over the entire AIS for each model)."

As mentioned in the comment, there are large discrepancies between the models. Especially the MRI-CGCM3 model largely overestimates the AIS SMB compared to RACMO2 (+1320 Gt year $^{-1}$, see the figure below).

[Figure]

**Figure S5.** Mean Antarctic Ice Sheet surface mass balance (Gt year[-1]) simulated by all the models used in this study.

*Figure 2:* You show the average while above you give the number for the median.

We have now replaced the median by the average.

*"who have shown that due to the lower spatial resolution of GCMs in comparison to the regional model, SMB is underestimated at the coasts while an overestimation occurs in the interior of the ice sheet."*
*Resolution might play a role but model's physics also plays a major role. E.g. Fig S1 shows that MRI-CGCM3 and ECHAM-wiso have much large SMB at the margins than RACMO2, whereas they have a lower resolution.*

Thank for your remark. We have added this sentence in the text:

However, models with similar resolutions may also have very different results, in particular in coastal regions (relative SMB biases of +47% and +100% for CCSM4 and MRI-CGCM3 respectively compared to RACMO for DML coast over the 1979-2005 period), suggesting a critical role of model physics in some of the GCM biases.

*Fig. S3:* Add the isotope-enabled models

As suggested, the isotope-enabled models have been added on the figure.

*"confirming that the spatial resolution has a crucial impact on the simulated SMB."*
*This is not convincing and not the dominant factor in my point of view.*

We have added a new sentence on the role of the model's physics in the new version of the manuscript (see the previous answer on the same topic).

*P11*
*"According to these reconstructions, this sensitivity has increased a lot for the recent period (1950–2005; 15.52 Do you think it is realistic? I don't find such an increase in sensitivity in Frieler et al. (2015)?*

We totally agree that this large increase in the SMB sensitivity to surface temperature using these reconstructions is quite surprising. Actually, as you mentioned, Frieler et al. (2015) do not obtain

such an increase. This could suggest that the reconstructions used in this study suffer from issues. We have added a sentence accordingly to this result:

"However, Frieler et al. (2015) do not obtain such an increase in SMB sensitivity (only ~+40%)."

*Figure 6:* I don't understand why for WAIS and AP, 'reconstructions' (black line) is lower than model mean, while for the combination of both (West Antarctica), 'reconstructions' is larger than the model mean? + typos in the legend.

The sensitivity factor for West Antarctica is not the average of the sensitivity factors of AP and WAIS. For the three aggregated regions (i.e. West Antarctica, East Antarctica, and Antarctica), our resulting sensitivity factors are based on SMB and SAT averaged over the regions. Because of some compensations between regions, what is observed for AP and WAIS can be different from what is observed for West Antarctica. The same behavior is noticed for Antarctica as a whole. Sensitivity factors deduced from the reconstruction for all sub-Antarctic regions are lower than the model mean, while for the continent as a whole, the value for the reconstruction is very close to the model mean.

*P12*
*"The analysis of isotope-enabled model results reinforces this hypothesis (Fig. 7): the iHadCM3 outputs show high correlations between these two variables."*
*In the sub-section 4.3, you only focus on the iHadCM3 outputs without explicitly announcing it and explaining why you did this choice.*

Throughout the text we mainly focused on the iHadCM3 model because, in contrast to the other isotope-enabled models (ECHAM5-wiso and ECHAM5/MPI-OM), iHadCM3 offers an ensemble of simulations, which is a significant advantage for data assimilation.

We added a few words on the reason of our choice at the end of the section 3.1.:

"Because iHadCM3 offers an ensemble of seven simulations, while the other isotope-enable models have only a single realization, we mainly focus on the iHadCM3 outputs in the manuscript. Dealing with an ensemble instead of a single simulation increases the probability of finding model results close to the assimilated records during the data assimilation process."

*P16*
*"(estimated by the weighted variance of the particles with non-zero weight)"*
*Define this weight/metric in the method section. What is the threshold?*

After each particle has received a weight depending on its likelihood, all the weights are multiplied by the total particle number. Then, the weights are rounded to the nearest integer toward negative infinity. Therefore, the maximum value of the weight is the number of particles and the minimum value is zero. We have specified in the new version of the manuscript how the weights are computed:

"Depending on its likelihood, each particle receives a weight. Then, all the weights are multiplied by the number of particles and rounded to the nearest integer toward negative infinity by ensuring that the sum of the weights equals the number of particles throughout the data assimilation process (see Dubinkina et al., 2011 for details)."

*"When assimilating both δ18O and SMB, the SMB reconstruction is in good agreement with the reconstruction of Thomas et al. (2017)."*

*As expected as Thomas is assimilated.*

Indeed, this is expected. However, we assimilate both $\delta^{18}O$ and SMB and not only SMB. Therefore, we constrain the model with two types of information. This can lead to a SMB reconstruction different from the reconstruction of Thomas et al. (2017) and indeed the reconstruction is different than the one assimilating only SMB (Figure S8). Additionally, if model outputs and assimilated records are too different, the resulting data assimilation-based reconstruction can highly differ from the data assimilated. If the resulting data assimilation-based reconstruction is close to the assimilated records, it means that no inconsistency is found between model results and the assimilate records.

Nevertheless, as this is not a surprising result, we have added "as expected" at the end of the sentence.

*P18*
*"who suggest an increase of the SMB sensitivity to surface temperature for the future in Antarctica,"*
*Can you give a number?*

According to Frierler et al. (2015), this increase is about 40% (Table 1). It has been added in the new version of the manuscript.

*"The GCMs may have biases in the simulated temperature changes or in their response to anthropogenic forcing."*
*This is very general, what are the known biases in GCMs?*

We agree that this sentence in the discussion/conclusions section is very general. We have added a couple of sentences regarding the GCM biases:

"The GCMs may have biases in the simulated temperature changes. For example, as shown by Klein et al. (2019), GCMs display on average a homogeneous warming over Antarctica during the last decades while observations mainly show a warming for West Antarctica with no significant change for East Antarctica. Additionally, climate model simulations generally display a warming starting in the 19$^{th}$ century in Antarctica while it begins much later in proxy-based reconstructions (Abram et al., 2016)."

*"This may contribute to an overestimation of the contribution of the simple thermodynamic link between temperature and precipitation and thus snow accumulation while it underestimates the role of changes in atmospheric circulation variability.*
*"Any reference on this point?*

We have added three papers supporting this point.

1. Abram, N. J., McGregor, H. V., Tierney, J. E., Evans, M. N., McKay, N. P., Kaufman, D. S., Thirumalai, K., Martrat, B., Goosse, H., Phipps,S. J., Steig, E. J., Kilbourne, K. H., Saenger, C. P., Zinke, J., Leduc, G., Addison, J. A., Mortyn, P. G., Seidenkrantz, M. S., Sicre, M. A.,Selvaraj, K., Filipsson, H. L., Neukom, R., Gergis, J., Curran, M. A., and Von Gunten, L. (2016): Early onset of industrial-era warming across the oceans and continents, Nature, 536, 411–418, https://doi.org/10.1038/nature19082.

2. Klein, F., Abram, N. J., Curran, M. A. J., Goosse, H., Goursaud, S., Masson-Delmotte, V., Moy, A., Neukom, R., Orsi, A., Sjolte, J., Steiger, N., Stenni, B., and Werner, M. (2019): Assessing the

robustness of Antarctic temperature reconstructions over the past 2 millennia using pseudoproxy and data assimilation experiments, Clim. Past, 15, 661–684, https://doi.org/10.5194/cp-15-661-2019.

3. PAGES 2k-PMIP3 group: Continental-scale temperature variability in PMIP3 simulations and PAGES 2k regional temperature reconstructions over the past millennium (2015), Clim. Past, 11, 1673–1699, https://doi.org/10.5194/cp-11-1673-2015.

The first paper shows that GCMs may imperfectly simulate the main mode of atmospheric variability over the last millennium. The other papers suggest that the model response to anthropogenic forcing (radiative forcing) is too important relatively to changes in general atmospheric circulation.

*"According to Neukom et al. (2018), uncertainties in the reconstructions (the noise in proxy data and the deficiencies in the reconstruction methods) and the data sampling could be an explanation of the observed discrepancy between models and reconstructions."*
*Give some key details on how it is proven.*

We have added the method used by Neukom et al. (2018) in the new version of the manuscript:

"To understand the potential origin of the disagreements between model results and reconstructions over the last millennium, Neukom et al. (2018) used pseudoproxy experiments. They found that uncertainties in the reconstructions (the noise in proxy data and the properties of the reconstruction methods) and the data sampling could be an explanation for many observed discrepancies between models and reconstructions."

*"surface temperature over the period 1958–2010"*
*Add the reference (Nicolas and Bromwich, 2014)*

Done.

*P19*
*"Regarding changes in SMB over the last two centuries, our reconstruction shows large regional differences in SMB trends, both in magnitude and in sign, in accordance with Medley and Thomas (2019; Fig. S12)."*
*A word on the fact that DA assimilate Thomas 2017, which use the same ice core dataset as in Medley and Thomas 2019? So it is not surprising that patterns are similar?*

As the method used by Medley and Thomas (2019) is different than ours, we could have had different results (even if the ice core dataset is the same). Unlike their method, we do not make any assumption on the stationarity of the link between the reanalysis (that they use) and the ice core dataset. Getting similar results thus shows that by using different methods, we obtain similar results, which gives more robustness to these results. However, we have added something in the corresponding sentence accordingly:

"Regarding changes in SMB over the last two centuries, our reconstruction shows large regional differences in SMB trends, both in magnitude and in sign, in accordance with Medley and Thomas (2019; Fig. S12) who used the same ice core dataset but a different method."

*"This is supported by a strong link between these two variables in observations, in particular for East Antarctica (r=0.82, statistically significant)."*
*Specify that is between Thomas et al 2017 and NB14, and does not work with Stenni2017*

The specification has been added in the text:

"This is supported by a strong link between these two variables in observations when using snow accumulation data from Thomas et al. (2017) and surface temperatures from Nicolas and Bromwich (2014), in particular for East Antarctica (r=0.82, statistically significant)."

*"By using data assimilation, no assumption such as stationarity or long calibration periods is required to estimate the link between variables"*
*Please also include the limitations of the data assimilation method*

We propose to add this sentence:

[revised manuscript text omitted]

~~Antarctic Ice Sheet Surface Mass Balance mm w.e. y$^{-1}$over 1979–2005 CE averaged over all the GCMs simulations (see Tab. A1 for the list) (top left), for RACMO2 (van Wessem et al., 2018) (top right), the difference between them (bottom left) and the distribution of the SMB simulated by RACMO2 and the GCMs as a function of elevation, binned in 400m elevation intervals (bottom right). The bars represent one standard deviation of the cell grids within each elevation bin. The equivalent of the latter panel for each model is provided on Fig. S10.~~

~~However, Figure S5 shows that the GCMs, compared to RACMO2, underestimate SMB in areas below 1500 m (mean bias of -55 mm w.e. year$^{-1}$; relative bias: -15%) over 1979–2005. For the areas above 1500 m, the mean bias of the simulated SMB by GCMs compared to RACMO2 is 11 mm w.e. year$^{-1}$ (relative bias: 11%). These results are in agreement with previous studies (e.g. Palerme et al., 2017; Genthon et al., 2009; Krinner et al., 2008) who have shown that due to the lower spatial resolution of GCMs in comparison to the regional model, SMB is underestimated at the coasts while an overestimation occurs in the interior of the ice sheet. The bias in the difference between the coastal and higher elevation regions are smaller for the models that have a higher spatial resolution, such as CCSM4 (Fig. S10) , confirming that the spatial resolution has a crucial impact on the simulated SMB~~ 
[revised manuscript text omitted]
 Sasgen, I.: Limits in detecting acceleration of ice sheet mass loss due to climate variability, Nature Geoscience, 6, 613–616, https://doi.org/10.1038/ngeo1874, http://dx.doi.org/10.1038/ngeo1874, 2013.

Wu, T., Song, L., Li, W., Wang, Z., Zhang, H., Xin, X., Zhang, Y., Zhang, L., Wu, F., Y, L., Zhang, F., Shi, X., Chu, M., Zhang, J., Fang, Y., Wang, F., Lu, Y., Liu, X., Wei, M., Liu, Q., Zhou, W., Dong, M., Zhao, Q., Ji, J., Laurent, L., and M, Z.: An overview of BCC climate system model development and application for climate change studies, Journal of Meteorological Research, 28, 34–56, https://doi.org/10.1007/s13351-014-3041-7.Supported, 2014.

Yukimoto, S., Adachi, Y., Hosaka, M., Sakami, T., Yoshimura, H., Hirabara, M., Tanaka, T. Y., Shindo, E., Tsujino, H., Deushi, M., Mizuta, R., Yabu, S., Obata, A., Nakano, H., Tsuyoshi, K., Ose, T., and Kitoh, A.: A New Global Climate Model of the Meteorological Research Institute: MRI-CGCM3 – Model Description and Basic Performance –, Journal of the Meteorological Society of Japan, 90A, 23–64, https://doi.org/10.2151/jmsj.2012-A02, http://japanlinkcenter.org/DN/JST.JSTAGE/jmsj/2012-A02?lang=en{&}from=CrossRef{&}type=abstract, 2012.

Antarctic Ice Sheet surface mass balance mm w.e. y$^{-1}$ for all the models used in this study over the 1979–2005 period.

Mean Antarctic Ice Sheet surface mass balance (Gt year$^{-1}$) simulated by all the models used in this study.

Distribution of the surface mass balance simulated by each GCM used in this study as a function of elevation, binned in 400m elevation intervals. The bars represent one standard deviation of the cell grids within each elevation bin.

5    Surface mass balance anomalies Gt y$^{-1}$ simulated by the GCMs (the average of all the available simulations has been represented; Tab. A1) and snow accumulation reconstructions (Thomas et al., 2017) for 1000–2005 and for 1800–2005 for all the Antarctic subregions. Anomalies are computed for the period 1800–2000. The shaded area corresponds to the range of the CESM1-CAM5 simulations. For visibility, data has been smoothed with a 100 year moving average for the last millennium and a 30 years moving average for the last 200 years.

10    Surface mass balance trends (in Gt 100y$^{-2}$) for West Antarctica, East Antarctica and Antarctica as a whole in GCMs, in isotopic climate models (ECHAM5-wiso, ECHAM5/MPIOM and HadCM3) and in reconstructions based on ice cores (Thomas et al., 2017) over 1950–2000. The number in brackets is the number of simulations. The trend computation is based on yearly data.

Annual correlations (r) between surface mass balance and surface temperature for all seven Antarctic regions (see Fig. 1 for
15 geographical definitions) for all the GCMs over the 1850–2005 AD. "1" is for the Plateau and "7" for DML Coast.

5 year correlations between SMB and $\delta^{18}$O, surface temperature and $\delta^{18}$O and, SMB and surface temperature for the seven Antarctic regions over 1870–199 time period from the ECHAM5-wiso outputs. In black, the correlation between the SMB reconstructions from Thomas et al. (2017) and the $\delta^{18}$O of the Antarctic ice cores aggregated for the seven Antarctic regions (Stenni et al., 2017).

20    5 year correlations between SMB and $\delta^{18}$O, surface temperature and $\delta^{18}$O and, SMB and surface temperature for the seven Antarctic regions over 1850–1995 from the ECHAM5/MPI-OM outputs. In black, the correlation between the SMB reconstructions from Thomas et al. (2017) and the $\delta^{18}$O of Antarctic ice cores aggregated for the seven Antarctic regions (Stenni et al., 2017).

Reconstructed surface temperatures (5-year mean) for West Antarctica, East Antarctica and Antarctica as a whole from
25 our data assimilation experiment using the ECHAM5-wiso outputs and, $\delta$018 (Stenni et al., 2017) and SMB reconstruction (Thomas et al., 2017) as data. The period is 1800–2010. *DA $\delta^{18}$O* is the data assimilation experiment using only the $\delta^{18}$O data to constrain the model while *DA SMB* uses only the SMB reconstruction and *DA $\delta^{18}$O and SMB* uses both. For each experiment and each region, the correlation (*r*) between the reconstruction based on ice cores and that based on data assimilation is computed. The shaded areas represent $\pm$ 1 standard deviation of the model particles.

30    Reconstructed surface temperatures (5-year mean) for West Antarctica, East Antarctica and Antarctica as a whole from data assimilation experiment using the ECHAM5-MPI/OM outputs and, $\delta$018 (Stenni et al., 2017) and SMB reconstruction (Thomas et al., 2017) as data. The period is 1800–2010. *DA $\delta^{18}$O* is the data assimilation experiment using only the $\delta^{18}$O data to constrain the model while *DA SMB* uses only the SMB reconstruction and *DA $\delta^{18}$O and SMB* uses both. For each experiment and each region, the correlation (*r*) between the reconstruction based on ice cores and that based on data assimilation is
35 computed. The shaded areas represent $\pm$ 1 standard deviation of the model particles.

Slopes (°C 100yr⁻¹) of each surface temperature reconstruction (Stenni et al., 2017; Klein et al., 2019; Nicolas and Bromwich, 2014; in this study) over the 1961–2010 period for West Antarctica, East Antarctica and the Antarctica. Statistically significant (p-value < 0.05) trends are represented by a star.

5-year mean correlations between the three surface temperature reconstructions from data assimilation experiments using the ECHAM5-MPI/OM outputs, ECHAM5-wiso outputs and the iHadCM3 outputs, and the surface temperature reconstructions from Nicolas and Bromwich (2014) over the 1958–2010 for the East, West and the whole Antarctica.

5-year mean correlations between the three surface temperature reconstructions from data assimilation experiments using the iHadCM3 outputs and the statistical reconstruction of Stenni et al. (2017), with the surface temperature reconstructions from Nicolas and Bromwich (2014) over the 1958–2010 period for East Antarctica, West Antartica and Antarctica as a whole. All the correlations are performed on detrended time series. Stars represent statistically significant correlations (p-value<0.10).

Reconstructed SMB (5-year mean) for West Antarctica, East Antarctica and Antarctica as a whole from data assimilation experiment using the ECHAM5-wiso outputs and, δ018 (Stenni et al., 2017) and SMB reconstruction (Thomas et al., 2017) as data. The period is 1800–2010. *DA δ¹⁸O* is the data assimilation experiment using only the δ¹⁸O data to constrain the model while *DA SMB* uses only the SMB reconstruction and *DA δ¹⁸O and SMB* uses both. For each experiment and each region, the correlation (*r*) between the reconstruction based on ice cores and that based on data assimilation is computed. The shaded areas represent ± 1 standard deviation of the model particles.

Reconstructed SMB (5-year mean) for West Antarctica, East Antarctica and Antarctica as a whole from data assimilation experiment using the ECHAM5-MPI/OM outputs and, δ018 (Stenni et al., 2017) and SMB reconstruction (Thomas et al., 2017) as data. The period is 1800–2010. *DA δ¹⁸O* is the data assimilation experiment using only the δ¹⁸O data to constrain the model while *DA SMB* uses only the SMB reconstruction and *DA δ¹⁸O and SMB* uses both. For each experiment and each region, the correlation (*r*) between the reconstruction based on ice cores and that based on data assimilation is computed. The shaded areas represent ± 1 standard deviation of the model particles.

Spatial Antarctic surface mass balance trends (mm w.e. y⁻¹ decade⁻¹) over the 1801–2000, 1957–2000 and 1979–2000 periods from 1) our data assimilation-based reconstruction using the iHadCM3 outputs constrained by both δ¹⁸O and SMB (first row) and from 2) Medley and Thomas (2019; second row).

**S1: Statistical surface temperature reconstructions from Stenni et al. (2017)**

 Using the δ¹⁸O composites, Stenni et al. (2017) reconstructed regional surface temperature over the last two millennia based on the statistical relationship between δ¹⁸O and surface temperature. Three methods have been used to scale the δ¹⁸O composites. In the first approach, the regional slopes between δ¹⁸O and temperatures were computed from the outputs of the ECHAM5-wiso model forced by ERA-Interim atmospheric reanalysis (Goursaud et al., 2018) over the 1979–2013 period. In the second method, the reconstruction obtained from the first method for the WAIS region is corrected using an independent temperature record: the borehole temperature reconstruction at WAIS divide (Orsi et al., 2012). This allows to match the cooling trend over the 1000–1600 period (Stenni et al., 2017). This method provides a different reconstruction for the WAIS

region – implying thus also the West Antarctic and the whole Antarctic reconstructions –, but not for the regions in East Antarctica. Finally, in the third method, the regional normalized $\delta^{18}$O composites have been scaled to the variance of the surface temperature observations (e.g. Nicolas and Bromwich, 2014) over the 1960–1990 period. The second reconstruction is used throughout this study for two reasons: 1) the third method is based on the surface temperature observations, which are

5   used here to estimate the skill of the reconstructions which could lead to a bias; 2) the correction introduced in the second method is expected to improve the reconstruction compared to the previous method. The temporal resolution of these surface temperature reconstruction is the same as the $\delta^{18}$O composites: 10 years for 0–1800 period and 5 years for 1800–2010 period.

**S2: Defining uncertainties associated with proxy data used during data assimilation process**

Data assimilation requires estimates of the uncertainty associated with the proxy data used. Unfortunately, uncertainty estima-

10   tions are not provided with  published reconstructions used here and the instrumental time series are too short to reliably derive the uncertainty. If we apply the same error for all the Antarctic regions, the assimilation will tend to give more weight to the time series that have more variance (i.e. the high-accumulation regions). On the other hand, if we apply an error proportional to the standard deviation of the time series, each region will tend to have the same weight. The uncertainly could also be related to the  amount of ice cores included in each regional composite, but the

15   link between this number and the quality of the composite is not straightforward (Stenni et al., 2017). Several experiments have been performed to test the impact of different estimates of the data uncertainties on the data assimilation results. The results are qualitatively similar  to standard choices of the uncertainty (Klein et al., 2019). The experiments shown here assume a signal to noise ratio of 1 for each regional composite. This is probably an optimistic estimate but this has the advantage of providing a strong data constraint and the comparison of the reconstruction using data assimilation with instrumental data indicates a good

20   skill of the methods using this value.

**S3: Present-day AIS SMB simulated by GCMs**

Overall, the AIS SMB simulated by GCMs is in good agreement with the SMB simulated by the regional climate model RACMO2 over the last decades (1979–2005, $R^2 = 0.63$; Figs. S5, S6, S7 and S8). As expected, both display high values of SMB along the coast (>300 mm w.e. year$^{-1}$) – especially for West Antarctica and the Antarctic Peninsula – and lower

25   values at high elevations (e.g. the Plateau: <100 mm w.e. year$^{-1}$). The mean of the SMB over the entire AIS simulated by the selected models (including isotope-enable models) is 6.4 mm w.e. year$^{-1}$ lower than the SMB simulated by RACMO2 over the 1979–2005 period (relative bias: -3.4%; Fig. S8 for the correlation plots for each model and Fig. S9 for the integrated SMB over the entire AIS for each model).

However, Figure S5 shows that the GCMs, compared to RACMO2, underestimate SMB in areas below 1500 m (mean bias of

30   -55 mm w.e. year$^{-1}$; relative bias: -15%) over 1979–2005. For the areas above 1500 m, the mean bias of the simulated SMB by GCMs compared to RACMO2 is 11 mm w.e. year$^{-1}$ (relative bias: 11%). These results are in agreement with previous studies

(e.g. Palerme et al., 2017; Genthon et al., 2009; Krinner et al., 2008) who have shown that due to the lower spatial resolution of GCMs in comparison to the regional model, SMB is underestimated at the coasts while an overestimation occurs in the interior of the ice sheet. The bias in the difference between the coastal and higher elevation regions are smaller for the models that have a higher spatial resolution, such as CCSM4 (Fig. S10), confirming that the spatial resolution has a crucial impact on the simulated SMB. However, models with similar resolutions may also have very different results, in particular in coastal regions (relative SMB biases of +47% and +100% for CCSM4 and MRI-CGCM3 respectively compared to RACMO for DML coast over the 1979–2005 period), suggesting a critical role of model physics in some of the GCM biases.

[Figure]

**Figure S1.** Surface mass balance anomalies [Gt y$^{-1}$] simulated by the GCMs (the average of all the available simulations has been represented; Tab. A1) and snow accumulation reconstructions (Thomas et al., 2017) for 1000–2005 and for 1800–2005 for all the Antarctic subregions. Anomalies are computed for the 1800–2000 period. The shaded area corresponds to the range of the CESM1-CAM5 simulations. For visibility, data has been smoothed with a 100 year moving average for the last millennium and a 30 years moving average for the last 200 years.

[Figure]

**Figure S2.** Annual correlations (r) between surface mass balance and surface temperature for all seven Antarctic regions (see Fig. 1 for geographical definitions) for all the GCMs over the 1850–2005 period.

[Figure]

**Figure S3.** 5-year mean correlations between surface temperature and $\delta^{18}$O (blue) and between SMB and surface temperature (green) for the seven Antarctic regions for the entire period simulation (1871–2010 for ECHAM5-wiso and 801–2000 for ECHAM5/MPI-OM).

[Figure]

**Figure S4.** Spatial Antarctic surface mass balance trends (mm w.e. y$^{-1}$ decade$^{-1}$) over the 1801–2000, 1957–2000 and 1979–2000 periods from 1) our data assimilation-based reconstruction using the iHadCM3 outputs constrained by both $\delta^{18}$O and SMB (first row) and from 2) Medley and Thomas (2019; second row).

[Figure]

**Figure S5.** Antarctic Ice Sheet Surface Mass Balance [mm w.e. y$^{-1}$] over 1979–2005 CE averaged over all the GCMs simulations (see Tab. A1 for the list) (top left), for RACMO2 (van Wessem et al., 2018) (top right), the difference between them (bottom left) and the distribution of the SMB simulated by RACMO2 and the GCMs as a function of elevation, binned in 400m elevation intervals (bottom right). The bars represent one standard deviation of the cell grids within each elevation bin. The equivalent of the latter panel for each model is provided on Fig. S10.

[Figure]

**Figure S6.** Correlation plot of SMB climatology from GCM mean (average over all the GCMs including isotope-enabled models) as a function of RACMO SMB over the 1979–2005 period. $R^2$ is the determination coefficient and bias the average of the difference between GCM mean and RACMO (in mm w.e. year$^{-1}$). Red (blue) dots are for places where the altitude is lower (higher) than 1500m. See Fig. S8 for each model.

[Figure]

**Figure S7.** Antarctic Ice Sheet surface mass balance [mm w.e. y⁻¹] for all the models used in this study over the 1979–2005 period.

[Figure]

**Figure S8.** As in Fig. S6 but for all GCMs.

[Figure]

**Figure S9.** Mean Antarctic Ice Sheet surface mass balance (Gt year$^{-1}$) simulated by all the models used in this study.

[Figure]

**Figure S10.** Distribution of the surface mass balance simulated by all climate models used in this study as a function of elevation, binned in 400m elevation intervals. The bars represent one standard deviation of the cell grids within each elevation bin.

[Figure]

**Figure S11.** Reconstructed surface temperatures (5-year mean) for West Antarctica, East Antarctica and Antarctica as a whole from our data assimilation experiment using the ECHAM5-wiso outputs and, $\delta^{18}$O (Stenni et al., 2017) and SMB reconstruction (Thomas et al., 2017) as data. The period is 1800–2010. *DA $\delta^{18}$O* is the data assimilation experiment using only the $\delta^{18}$O data to constrain the model while *DA SMB* uses only the SMB reconstruction and *DA $\delta^{18}$O and SMB* uses both. For each experiment and each region, the correlation (*r*) between the reconstruction based on ice cores and that based on data assimilation is computed. The shaded areas represent $\pm$ 1 standard deviation of the model particles.

[Figure]

**Figure S12.** Reconstructed surface temperatures (5-year mean) for West Antarctica, East Antarctica and Antarctica as a whole from data assimilation experiment using the ECHAM5-MPI/OM outputs and, $\delta^{18}$O (Stenni et al., 2017) and SMB reconstruction (Thomas et al., 2017) as data. The period is 1800–2010. *DA $\delta^{18}$O* is the data assimilation experiment using only the $\delta^{18}$O data to constrain the model while *DA SMB* uses only the SMB reconstruction and *DA $\delta^{18}$O and SMB* uses both. For each experiment and each region, the correlation (*r*) between the reconstruction based on ice cores and that based on data assimilation is computed. The shaded areas represent $\pm$ 1 standard deviation of the model particles.

[Figure]

**Figure S13.** Reconstructed SMB (5-year mean) for West Antarctica, East Antarctica and Antarctica as a whole from data assimilation experiment using the ECHAM5-wiso outputs and, $\delta^{18}$O (Stenni et al., 2017) and SMB reconstruction (Thomas et al., 2017) as data. The period is 1800–2010. *DA $\delta^{18}$O* is the data assimilation experiment using only the $\delta^{18}$O data to constrain the model while *DA SMB* uses only the SMB reconstruction and *DA $\delta^{18}$O and SMB* uses both. For each experiment and each region, the correlation ($r$) between the reconstruction based on ice cores and that based on data assimilation is computed. The shaded areas represent $\pm$ 1 standard deviation of the model particles.

[Figure]

**Figure S14.** Reconstructed SMB (5-year mean) for West Antarctica, East Antarctica and Antarctica as a whole from data assimilation experiment using the ECHAM5-MPI/OM outputs and, $\delta^{18}$O (Stenni et al., 2017) and SMB reconstruction (Thomas et al., 2017) as data. The period is 1800–2010. *DA $\delta^{18}$O* is the data assimilation experiment using only the $\delta^{18}$O data to constrain the model while *DA SMB* uses only the SMB reconstruction and *DA $\delta^{18}$O and SMB* uses both. For each experiment and each region, the correlation (*r*) between the reconstruction based on ice cores and that based on data assimilation is computed. The shaded areas represent ± 1 standard deviation of the model particles.

**Table S1.** Surface mass balance trends (in Gt 100y$^{-2}$) for West Antarctica, East Antarctica and Antarctica as a whole in GCMs, in isotopic climate models (ECHAM5-wiso, ECHAM5/MPIOM and HadCM3) and in reconstructions based on ice cores (Thomas et al., 2017) over 1950–2000. The number in brackets is the number of simulations. The trend computation is based on yearly data.

| | West Antarctica | | | East Antarctica | | | Antarctica | | |
|---|---|---|---|---|---|---|---|---|---|
| | min | max | mean | min | max | mean | min | max | mean |
| bcc-csm1-1 (3) | -29.46 | 152.47 | 77.62 | -63.11 | 381.09 | 200.39 | -92.57 | 533.56 | 278.01 |
| CCSM4 (6) | 148.02 | 390.50 | 234.13 | 274.32 | 455.65 | 368.19 | 469.24 | 846.15 | 602.32 |
| CSIRO-Mk3L-1-2 (1) | | | 3.14 | | | 135.86 | | | 139.00 |
| GISS-E2-R (6) | 25.69 | 183.66 | 107.27 | -71.92 | 250.18 | 140.43 | -46.23 | 416.21 | 247.71 |
| HadCM3 (10) | 4.79 | 150.27 | 70.75 | -68.85 | 242.18 | 89.39 | -34.18 | 303.11 | 160.14 |
| IPSL-CM5A-LR (6) | 57.07 | 123.78 | 99.07 | -104.18 | 66.82 | -10.06 | -47.11 | 174.30 | 89.01 |
| MPI-ESM-P (2) | -33.85 | -28.74 | -31.30 | 54.75 | 231.84 | 143.29 | 26.01 | 197.99 | 112.00 |
| MRI-CGCM3 (3) | 28.62 | 178.64 | 86.45 | -59.28 | 242.24 | 125.66 | -7.19 | 420.89 | 212.11 |
| CESM1-CAM5 (12) | 30.90 | 349.67 | 153.07 | 55.72 | 340.24 | 162.27 | 161.99 | 592.43 | 315.34 |
| iHadCM3 (6) | 76.23 | 232.69 | 162.29 | 15.52 | 350.87 | 213.61 | 115.85 | 542.61 | 375.90 |
| ECHAM5-wiso (1) | | | -8.79 | | | 195.22 | | | 186.43 |
| ECHAM5/MPIOM (1) | | | 41.44 | | | 35.43 | | | 76.87 |
| Reconstructions (1) | | | 256.74 | | | -35.80 | | | 220.95 |

**Table S2.** 5-year mean correlations between the three surface temperature reconstructions from data assimilation experiments using the iHadCM3 outputs and the statistical reconstruction of Stenni et al. (2017), with the surface temperature reconstructions from Nicolas and Bromwich (2014) over the 1958–2010 period for East Antarctica, West Antartica and Antarctica as a whole. All the correlations are performed on detrended time series. Stars represent statistically significant correlations (p-value<0.10).

| | West Antarctica | East Antarctica | Antarctica |
|---|---|---|---|
| DA $\delta^{18}$O | -0.02 | -0.16 | -0.25 |
| DA SMB | -0.19 | 0.51 | 0.31 |
| DA $\delta^{18}$O and SMB | 0 | 0.60[*] | 0.44 |
| Stenni et al. (2017) | 0.45[*] | -0.20 | 0.12[*] |

**Table S3.** Slopes (°C 100yr$^{-1}$) of each surface temperature reconstruction (Stenni et al., 2017; Klein et al., 2019; Nicolas and Bromwich, 2014 ; in this study) over the 1961–2010 period for West Antarctica, East Antarctica and the Antarctica. Statistically significant (p-value < 0.05) trends are represented by a star.

| Dataset | West Antarctica | East Antarctica | Antarctica |
|---|---|---|---|
| Stenni et al. (2017) | | | |
|     Stat ECHAMvariance | 1.69* | 0.75* | 1.27 |
|     Stat borehole | 2.07* | 0.75* | 0.77* |
| Klein et al. (2018) | | | |
|     DA ECHAM5-wiso | 1.15 | 0.94 | 0.98 |
|     DA ECHAM5/MPI-OM | 1.0 | 0.48 | 0.59 |
| Nicolas and Bromwich (2014) | | | |
| | 2.22* | 0.53 | 0.90* |
| In this study | | | |
|     DA $\delta^{18}$O and SMB iHadCM3 | 0.99* | 0.60* | 0.69* |

**Table S4.** 5-year mean correlations between the three surface temperature reconstructions from data assimilation experiments using the ECHAM5-MPI/OM outputs, ECHAM5-wiso outputs, the iHadCM3 outputs and the two surface temperature reconstructions of Stenni et al. (2017) with the surface temperature reconstruction from Nicolas and Bromwich (2014) over the 1958–2010 for East Antarctica, West Antarctica and Antarctica as a whole.

| | West Antarctica | | | East Antarctica | | | Antarctica | | |
|---|---|---|---|---|---|---|---|---|---|
| | ECHAM5-MPI/OM | ECHAM5-wiso | iHadCM3 | ECHAM5-MPI/OM | ECHAM5-wiso | iHadCM3 | ECHAM5-MPI/OM | ECHAM5-wiso | iHadCM3 |
| DA $\delta^{18}$O | 0.57 | 0.78 | 0.69 | 0.19 | 0.08 | 0.13 | 0.50 | 0.47 | 0.34 |
| DA SMB | 0.40 | 0.52 | 0.55 | 0.27 | 0.53 | 0.60 | 0.28 | 0.58 | 0.65 |
| DA $\delta^{18}$O and SMB | 0.53 | 0.65 | 0.72 | 0.34 | 0.48 | 0.61 | 0.59 | 0.71 | 0.73 |
| Stenni et al. (2017) | 0.79 | | | 0.10 | | | 0.57 | | |

---

## Author Response (AR2)

Dear authors,

The revised manuscript was reviewed by one referee who suggested to accept this article subject to minor revision. Please consider suggested changes and submit the revised manuscript together with the marked manuscript and the response letter.

I look forward seeing the revised manuscript.

Kenny Matsuoka
TC/TCD Editor

Dear Editor,

Thank you again for editing our manuscript. You can find below the point by point response to each reviewer's comment, along with the updated version of our manuscript with track changes.

On behalf of all co-authors,

Dalaiden Quentin

The Referee's comments below are in italics, our answer in plain font in blue

*The authors answered to all my comment, and I thank them for taking them into account.*
*I still have 3 minor and 1 major comments bellow.*

We would like to thank again the Referee for their careful evaluation and for all the suggestions that helped improve the manuscript.

*= minor =*
*"More specifically, with only a few uncertain data, it is expected that the reconstruction based on our data assimilation method may show less variance than reconstructions provided by some other methods (as observed previously; e.g. Goosse et al. 2010). Nevertheless, we did not discuss much this point in the manuscript as it critically depends on the uncertainty of the input data, that is itself not well known. »*

*Can you add a short sentence on this point in the text, maybe in the discussion?*

We accordingly have added a sentence regarding the variance of our reconstruction in the discussion section:

"Our reconstruction displays smaller variance in time than the reconstruction from Stenni et al. (2017), which is a standard characteristic of estimates based on data assimilation using only a few uncertain data (e.g. Goosse et al., 2010)."

*= minor =*
*In the text, at the end of Section 3.1:*

*« Considering all particles weights, we can compute a weighted average, providing a reconstruction for this time step. In this study, the ensemble members are derived from three climate model outputs: ECHAM5-MPI/OM (Sjolte et al., 2018), ECHAM5-wiso (Steiger et al., 2017) and iHadCM3 (Tindall et al., 2009; Holloway et al., 2016b). »*

*As models have already been presented with more details in Section 1.2, I suggest to change for something like:*
*« Considering all particles weights, we can compute a weighted average, providing a reconstruction for this time step. In this study, the ensemble members are derived from the three isotope-enabled climate model outputs ECHAM5-MPI/OM, ECHAM5-wiso and iHadCM3, presented at Section 1.2. »*

We have applied the change in the new version of the manuscript. Thank you for your suggestion.

*= major =*
*« Throughout the text we mainly focused on the iHadCM3 model because, in contrast to the other isotope-enabled models (ECHAM5-wiso and ECHAM5-MPI/OM), iHadCM3 offers an ensemble of simulations, which is a significant advantage for data assimilation.*

*We added a few words on the reason of our choice at the end of the section 3.1.:*
*"Because iHadCM3 offers an ensemble of seven simulations, while the other isotope-enable models have only a single realization, we mainly focus on the iHadCM3 outputs in the manuscript. Dealing with an ensemble instead of a single simulation increases the probability of finding model results close to the assimilated records during the data assimilation process." »*

*I computed the number of years by simulation:*

*ECHAM5-MPI/OM: 800–2000 = 1200 years*
*ECHAM5-wiso: 1871–2011 = 141 years*
*iHadCM3: seven x 1851-2003 = 7 x 153 years = 1071 years*

*If indeed ECHAM5-wiso gives a much lower number of particles than ECHAM5-MPI/OM and iHadCM3, you did not quantify how much the long simulation from ECHAM5-MPI/OM covers less internal variability than iHadCM3. I am not convinced it is necessarily true. The reason for which you chose to focus on iHadCM3 seems weak to me. One can wonder if it is related to the fact that iHadCM3 DA is the only reconstruction for which using both δ18O and SMB gives better results than δ18O alone for West Antarctica.*

*To answer to this comment, I suggest to include the results of the 3 models, or at least results of ECHAM5-MPI/OM together with iHadCM3, to draw your conclusions. I believe your results are robust, and the fact that it remains robust for the 3 models for East Antarctica and all-Antarctica is a strong assessment of it in my point of view (Table S4).*
*In particular I think Table S4 should replace Table 1 and be commented in the text, and Fig. 6 should be updated to include ECHAM5-MPI/OM.*

We agree that the potential number of particles is similar for iHadCM3 and ECHAM5-MPI/OM ensembles. However, the climate variability of the iHadCM3 ensemble is larger than the variability of the ECHAM5-MPI/OM simulation. For example, the regional variability in surface air temperature for the iHadCM3 are always larger than for the ECHAM5-MPI/OM ensemble (Table 1). We also generally observe a greater variance for the $\delta^{18}O$ and SMB variables for the iHadCM3 ensemble compared to the ECHAM5-MPI/OM ensemble (Tables 2 and 3). A wide climatic range increases the probability of finding model results close to the assimilated records and limits the risk of an overconfident reconstruction or of a degeneracy of the assimilation. Consequently, even though the ECHAM5-MPI/OM ensemble is slightly larger than the iHadCM3 ensemble, the ECHAM5-MPI/OM ensemble covers a more limited climatic range compared to the iHadCM3.

We first analysed the results from iHadCM3 because working with an ensemble allowed us to assess the uncertainty of the Antarctic SMB-SAT and $\delta^{18}O$-SAT relationships due to internal variability over the recent past (1850-2000) when comparing to observations and reconstructions based on proxies, which is not feasible with only one simulation such as the ECHAM5-MPI/OM simulation. Additionally, the iHadCM3 ensemble provides a more realistic Antarctic climate than the Antarctic climate provided by the ECHAM5-MPI/OM simulation:

1. Klein et al. (2019) have shown that the ECHAM5-MPI/OM simulation displays a surface warming during the 850-1850 period (+0.02 degree Celsius per century)

while other model simulations (7 different models) and reconstructions based on proxies show all a cooling (-0.03 and -0.06 degree Celsius per century, respectively; Figure 1 from the mentioned paper). Over the last decades, this simulation shows a cooling (-0.24 degree Celsius per century) while in models, reconstructions and observations, a warming is noticed for the Antarctic continent (1.54, 1.74 and 1.00 degree Celsius per century, respectively). iHadCM3 displays a warming from 0.28 to 3.10 degree Celsius per century for Antarctica as a whole over 1960-2000 AD.

2. According to our evaluation of the current AIS SMB, when comparing to the RACMO2 SMB, iHadCM3 shows a higher $R^2$ and a smaller bias than ECHAM5-MPI/OM (Figure S8).

Given that the manuscript already contains a lot of information, we think that adding figures from ECHAM5-MPI/OM in the main text will make the manuscript less readable. The discussion regarding the surface air temperature and snow accumulation reconstructions will contain much more numbers, making the reading difficult. Besides, as the Referee mentioned, our results are consistent between all the models and thus including these results will not change our main conclusions. If the reader would look at the results for the other models, all the results are in Supplementary Materials. We thus propose to only present figures from iHadCM3 in the main manuscript to avoid overloading the manuscript by information that only confirms our main findings. The figure for the other models are presented in the Supplementary Materials. Besides, as suggested by the Referee, we have replaced in the manuscript the Table 1 by the Table S4 including results from the three models with a discussion mainly focused on the results from iHadCM3.

Accordingly to the Referee's comment, we also have modified our initial paragraph discussing the reasons why we chose iHadCM3:

"Because iHadCM3 offers an ensemble of seven simulations, while the other isotope-enable models have only a single realization, we mainly focus on the iHadCM3 outputs in the manuscript. Dealing with an ensemble instead of a single simulation increases the probability of finding model results close to the assimilated records during the data assimilation process."

by this:

"Because iHadCM3 offers an ensemble of seven simulations, while the other isotope-enable models have only a single realization, we mainly focus on the iHadCM3 outputs in the main manuscript. The results from the other models are shown in the Supplementary Materials and provide similar conclusions to the ones obtained with iHadCM3. Using an ensemble allow us estimating the contribution of internal variability over the last century and the range provided by this ensemble is larger than the one given by the other two models, increasing the probability of finding model results close to the assimilated records during the data assimilation process. Additionally, iHadCM3 ensemble provides a more realistic Antarctic surface temperature and snow accumulation over recent past than the ECHAM5-MPI/OM simulation (Fig. S8; Klein et al., 2019)."

= *minor* =

*In Figure 7 vs. Fig. S11, the red shaded area show larger spread for iHadCM3 DA than for ECHAMwiso DA. This is logical as you show 1-sigma of model particles, and the number of particules is much lower in ECHAMwiso than in iHadCM3. But ideally the shaded area should reflect this, giving more uncertainty on reconstructions coming from ECHAMwiso, no? Can you compute a metric of the uncertainty of your DA reconstruction that decreases with increasing number of particle?*

Thank you for your comment. Indeed the variance of the reconstruction using iHadCM3 (Figure 7) is larger (0.47 degree Celsius for Antarctica as a whole when the two proxy types are assimilated) than for ECHAM5-wiso (0.30 degree Celsius respectively; Figure S11). We could think that dealing with a larger ensemble provides a more accurate reconstruction. However, the resulting uncertainty of the posterior (i.e. the final reconstruction) - computed as the standard deviation of the particles kept - depends on the prior (our data assimilation ensemble built from climate models outputs) and the observation network (here the proxies). Increasing the number of particles will not necessarily result in a more accurate reconstruction. A convergence is achieved after a critical number of particles. However, as the resulting variance of the DA reconstruction depends on the variance of the prior, if this prior underestimates the climate variability, it will fatally lead to an underestimation of the uncertainty of our reconstruction. Actually, the regional standard deviations of surface air temperature in the iHadCM3 ensemble are almost always larger than the variance in the ECHAM5-wiso ensemble (Table 1), which partly explains why we obtained a larger variability on our reconstruction based on iHadCM3. Furthermore, the data uncertainties have also a larger impact on the final uncertainty.

Table 1. Regional standard deviations of surface air temperature for each ensemble used in data assimilation (degree Celsius).

|  | ECHAM5-wiso (142 particles) | ECHAM5-MPI/OM (1201 particles) | iHadCM3 (1071 particles) |
|---|---|---|---|
| Plateau | 0.72 | 0.68 | 0.84 |
| Wilkes Land | 0.68 | 0.74 | 0.81 |
| Weddell | 0.96 | 0.94 | 0.96 |
| Peninsula | 0.64 | 0.78 | 0.88 |
| WAIS | 0.91 | 0.88 | 1.03 |
| Victoria Land | 0.89 | 0.69 | 0.84 |
| Dronning Maud Land | 0.65 | 0.64 | 0.76 |
| West Antarctica | 0.66 | 0.67 | 0.82 |
| East Antarctica | 0.66 | 0.64 | 0.78 |
| Antarctica | 0.58 | 0.58 | 0.72 |

Table 2. Regional standard deviations of $\delta^{18}O$ for each ensemble used in data assimilation (‰).

|  | ECHAM5-wiso (142 particles) | ECHAM5-MPI/OM (1201 particles) | iHadCM3 (1071 particles) |
|---|---|---|---|
| Plateau | 0.69 | 0.43 | 0.82 |
| Wilkes Land | 0.38 | 0.40 | 0.86 |
| Weddell | 0.81 | 0.69 | 1.2 |
| Peninsula | 0.46 | 0.51 | 0.88 |
| WAIS | 0.44 | 0.52 | 1.1 |
| Victoria Land | 0.75 | 0.74 | 1.4 |
| Dronning Maud Land | 0.53 | 0.65 | 0.71 |
| West Antarctica | 0.35 | 0.43 | 0.82 |
| East Antarctica | 0.56 | 0.38 | 0.74 |
| Antarctica | 0.47 | 0.35 | 0.68 |

Table 3. Regional standard deviations of SMB for each data assimilation ensemble (Gt year[-1]).

[revised manuscript text omitted]

**S1: Statistical surface temperature reconstructions from Stenni et al. (2017)**

Using the $\delta^{18}$O composites, Stenni et al. (2017) reconstructed regional surface temperature over the last two millennia based on the statistical relationship between $\delta^{18}$O and surface temperature. Three methods have been used to scale the $\delta^{18}$O composites. In the first approach, the regional slopes between $\delta^{18}$O and temperatures were computed from the outputs of the ECHAM5-wiso model forced by ERA-Interim atmospheric reanalysis (Goursaud et al., 2018) over the 1979–2013 period. In the second method, the reconstruction obtained from the first method for the WAIS region is corrected using an independent temperature record: the borehole temperature reconstruction at WAIS divide (Orsi et al., 2012). This allows to match the cooling trend over the 1000–1600 period (Stenni et al., 2017). This method provides a different reconstruction for the WAIS region – implying thus also the West Antarctic and the whole Antarctic reconstructions –, but not for the regions in East Antarctica. Finally, in the third method, the regional normalized $\delta^{18}$O composites have been scaled to the variance of the surface temperature observations (e.g. Nicolas and Bromwich, 2014) over the 1960–1990 period. The second reconstruction is used throughout this study for two reasons: 1) the third method is based on the surface temperature observations, which are used here to estimate the skill of the reconstructions which could lead to a bias; 2) the correction introduced in the second method is expected to improve the reconstruction compared to the previous method. The temporal resolution of these surface temperature reconstruction is the same as the $\delta^{18}$O composites: 10 years for 0–1800 period and 5 years for 1800–2010 period.

**S2: Defining uncertainties associated with proxy data used during data assimilation process**

Data assimilation requires estimates of the uncertainty associated with the proxy data used. Unfortunately, uncertainty estimations are not provided with published reconstructions used here and the instrumental time series are too short to reliably derive the uncertainty. If we apply the same error for all the Antarctic regions, the assimilation will tend to give more weight to the time series that have more variance (i.e. the high-accumulation regions). On the other hand, if we apply an error proportional to the standard deviation of the time series, each region will tend to have the same weight. The uncertainly could also be related to the amount of ice cores included in each regional composite, but the link between this number and the quality of the composite is not straightforward (Stenni et al., 2017). Several experiments have been performed to test the impact of different estimates of the data uncertainties on the data assimilation results. The results are qualitatively similar to standard choices of the uncertainty (Klein et al., 2019). The experiments shown here assume a signal to noise ratio of 1 for each regional composite. This is probably an optimistic estimate but this has the advantage of providing a strong data constraint and the comparison of the reconstruction using data assimilation with instrumental data indicates a good skill of the methods using this value.

**S3: Present-day AIS SMB simulated by GCMs**

Overall, the AIS SMB simulated by GCMs is in good agreement with the SMB simulated by the regional climate model RACMO2 over the last decades (1979–2005, $R^2$ = 0.63; Figs. S5, S6, S7 and S8). As expected, both display high values of SMB along the coast (>300 mm w.e. year$^{-1}$) – especially for West Antarctica and the Antarctic Peninsula – and lower values

at high elevations (e.g. the Plateau: <100 mm w.e. year$^{-1}$). The mean of the SMB over the entire AIS simulated by the selected models (including isotope-enable models) is 6.4 mm w.e. year$^{-1}$ lower than the SMB simulated by RACMO2 over the 1979–2005 period (relative bias: -3.4%; Fig. S8 for the correlation plots for each model and Fig. S9 for the integrated SMB over the entire AIS for each model).

5     However, Figure S5 shows that the GCMs, compared to RACMO2, underestimate SMB in areas below 1500 m (mean bias of -55 mm w.e. year$^{-1}$; relative bias: -15%) over 1979–2005. For the areas above 1500 m, the mean bias of the simulated SMB by GCMs compared to RACMO2 is 11 mm w.e. year$^{-1}$ (relative bias: 11%). These results are in agreement with previous studies (e.g. Palerme et al., 2017; Genthon et al., 2009; Krinner et al., 2008) who have shown that due to the lower spatial resolution of GCMs in comparison to the regional model, SMB is underestimated at the coasts while an overestimation occurs in the interior

10   of the ice sheet. The bias in the difference between the coastal and higher elevation regions are smaller for the models that have a higher spatial resolution, such as CCSM4 (Fig. S10), confirming that the spatial resolution has a crucial impact on the simulated SMB. However, models with similar resolutions may also have very different results, in particular in coastal regions (relative SMB biases of +47% and +100% for CCSM4 and MRI-CGCM3 respectively compared to RACMO for DML coast over the 1979–2005 period), suggesting a critical role of model physics in some of the GCM biases.

[Figure]

**Figure S1.** Surface mass balance anomalies [Gt y$^{-1}$] simulated by the GCMs (the average of all the available simulations has been represented; Tab. A1) and snow accumulation reconstructions (Thomas et al., 2017) for 1000–2005 and for 1800–2005 for all the Antarctic subregions. Anomalies are computed for the 1800–2000 period. The shaded area corresponds to the range of the CESM1-CAM5 simulations. For visibility, data has been smoothed with a 100 year moving average for the last millennium and a 30 years moving average for the last 200 years.

[Figure]

**Figure S2.** Annual correlations (r) between surface mass balance and surface temperature for all seven Antarctic regions (see Fig. 1 for geographical definitions) for all the GCMs over the 1850–2005 period.

[Figure]

**Figure S3.** 5-year mean correlations between surface temperature and $\delta^{18}$O (blue) and between SMB and surface temperature (green) for the seven Antarctic regions for the entire period simulation (1871–2010 for ECHAM5-wiso and 801–2000 for ECHAM5/MPI-OM).

[Figure]

**Figure S4.** Spatial Antarctic surface mass balance trends (mm w.e. y⁻¹ decade⁻¹) over the 1801–2000, 1957–2000 and 1979–2000 periods from 1) our data assimilation-based reconstruction using the iHadCM3 outputs constrained by both $\delta^{18}O$ and SMB (first row) and from 2) Medley and Thomas (2019; second row).

[Figure]

**Figure S5.** Antarctic Ice Sheet Surface Mass Balance [mm w.e. y$^{-1}$] over 1979–2005 CE averaged over all the GCMs simulations (see Tab. A1 for the list) (top left), for RACMO2 (van Wessem et al., 2018) (top right), the difference between them (bottom left) and the distribution of the SMB simulated by RACMO2 and the GCMs as a function of elevation, binned in 400m elevation intervals (bottom right). The bars represent one standard deviation of the cell grids within each elevation bin. The equivalent of the latter panel for each model is provided on Fig. S10.

[Figure]

**Figure S6.** Correlation plot of SMB climatology from GCM mean (average over all the GCMs including isotope-enabled models) as a function of RACMO SMB over the 1979–2005 period. $R^2$ is the determination coefficient and bias the average of the difference between GCM mean and RACMO (in mm w.e. year[-1]). Red (blue) dots are for places where the altitude is lower (higher) than 1500m. See Fig. S8 for each model.

[Figure]

**Figure S7.** Antarctic Ice Sheet surface mass balance [mm w.e. y$^{-1}$] for all the models used in this study over the 1979–2005 period.

[Figure]

**Figure S8.** As in Fig. S6 but for all GCMs.

[Figure]

**Figure S9.** Mean Antarctic Ice Sheet surface mass balance (Gt year[-1]) simulated by all the models used in this study.

[Figure]

**Figure S10.** Distribution of the surface mass balance simulated by all climate models used in this study as a function of elevation, binned in 400m elevation intervals. The bars represent one standard deviation of the cell grids within each elevation bin.

[Figure]

**Figure S11.** Reconstructed surface temperatures (5-year mean) for West Antarctica, East Antarctica and Antarctica as a whole from our data assimilation experiment using the ECHAM5-wiso outputs and, $\delta^{18}$0 (Stenni et al., 2017) and SMB reconstruction (Thomas et al., 2017) as data. The period is 1800–2010. *DA $\delta^{18}O$* is the data assimilation experiment using only the $\delta^{18}$O data to constrain the model while *DA SMB* uses only the SMB reconstruction and *DA $\delta^{18}O$ and SMB* uses both. For each experiment and each region, the correlation (*r*) between the reconstruction based on ice cores and that based on data assimilation is computed. The shaded areas represent $\pm$ 1 standard deviation of the model particles.

[Figure]

**Figure S12.** Reconstructed surface temperatures (5-year mean) for West Antarctica, East Antarctica and Antarctica as a whole from data assimilation experiment using the ECHAM5-MPI/OM outputs and, $\delta^{18}$O (Stenni et al., 2017) and SMB reconstruction (Thomas et al., 2017) as data. The period is 1800–2010. *DA $\delta^{18}$O* is the data assimilation experiment using only the $\delta^{18}$O data to constrain the model while *DA SMB* uses only the SMB reconstruction and *DA $\delta^{18}$O and SMB* uses both. For each experiment and each region, the correlation (*r*) between the reconstruction based on ice cores and that based on data assimilation is computed. The shaded areas represent ± 1 standard deviation of the model particles.

[Figure]

**Figure S13.** Reconstructed SMB (5-year mean) for West Antarctica, East Antarctica and Antarctica as a whole from data assimilation experiment using the ECHAM5-wiso outputs and, $\delta^{18}$O (Stenni et al., 2017) and SMB reconstruction (Thomas et al., 2017) as data. The period is 1800–2010. *DA $\delta^{18}$O* is the data assimilation experiment using only the $\delta^{18}$O data to constrain the model while *DA SMB* uses only the SMB reconstruction and *DA $\delta^{18}$O and SMB* uses both. For each experiment and each region, the correlation (*r*) between the reconstruction based on ice cores and that based on data assimilation is computed. The shaded areas represent $\pm$ 1 standard deviation of the model particles.

[Figure]

**Figure S14.** Reconstructed SMB (5-year mean) for West Antarctica, East Antarctica and Antarctica as a whole from data assimilation experiment using the ECHAM5-MPI/OM outputs and, $\delta^{18}$O (Stenni et al., 2017) and SMB reconstruction (Thomas et al., 2017) as data. The period is 1800–2010. *DA $\delta^{18}$O* is the data assimilation experiment using only the $\delta^{18}$O data to constrain the model while *DA SMB* uses only the SMB reconstruction and *DA $\delta^{18}$O and SMB* uses both. For each experiment and each region, the correlation (*r*) between the reconstruction based on ice cores and that based on data assimilation is computed. The shaded areas represent $\pm$ 1 standard deviation of the model particles.

**Table S1.** Surface mass balance trends (in Gt 100y$^{-2}$) for West Antarctica, East Antarctica and Antarctica as a whole in GCMs, in isotopic climate models (ECHAM5-wiso, ECHAM5/MPIOM and HadCM3) and in reconstructions based on ice cores (Thomas et al., 2017) over 1950–2000. The number in brackets is the number of simulations. The trend computation is based on yearly data.

| | West Antarctica | | | East Antarctica | | | Antarctica | | |
|---|---|---|---|---|---|---|---|---|---|
| | min | max | mean | min | max | mean | min | max | mean |
| bcc-csm1-1 (3) | -29.46 | 152.47 | 77.62 | -63.11 | 381.09 | 200.39 | -92.57 | 533.56 | 278.01 |
| CCSM4 (6) | 148.02 | 390.50 | 234.13 | 274.32 | 455.65 | 368.19 | 469.24 | 846.15 | 602.32 |
| CSIRO-Mk3L-1-2 (1) | | | 3.14 | | | 135.86 | | | 139.00 |
| GISS-E2-R (6) | 25.69 | 183.66 | 107.27 | -71.92 | 250.18 | 140.43 | -46.23 | 416.21 | 247.71 |
| HadCM3 (10) | 4.79 | 150.27 | 70.75 | -68.85 | 242.18 | 89.39 | -34.18 | 303.11 | 160.14 |
| IPSL-CM5A-LR (6) | 57.07 | 123.78 | 99.07 | -104.18 | 66.82 | -10.06 | -47.11 | 174.30 | 89.01 |
| MPI-ESM-P (2) | -33.85 | -28.74 | -31.30 | 54.75 | 231.84 | 143.29 | 26.01 | 197.99 | 112.00 |
| MRI-CGCM3 (3) | 28.62 | 178.64 | 86.45 | -59.28 | 242.24 | 125.66 | -7.19 | 420.89 | 212.11 |
| CESM1-CAM5 (12) | 30.90 | 349.67 | 153.07 | 55.72 | 340.24 | 162.27 | 161.99 | 592.43 | 315.34 |
| iHadCM3 (6) | 76.23 | 232.69 | 162.29 | 15.52 | 350.87 | 213.61 | 115.85 | 542.61 | 375.90 |
| ECHAM5-wiso (1) | | | -8.79 | | | 195.22 | | | 186.43 |
| ECHAM5/MPIOM (1) | | | 41.44 | | | 35.43 | | | 76.87 |
| Reconstructions (1) | | | 256.74 | | | -35.80 | | | 220.95 |

**Table S2.** 5-year mean correlations between the three surface temperature reconstructions from data assimilation experiments using the iHadCM3 outputs and the statistical reconstruction of Stenni et al. (2017), with the surface temperature reconstructions from Nicolas and Bromwich (2014) over the 1958–2010 period for East Antarctica, West Antartica and Antarctica as a whole. All the correlations are performed on detrended time series. Stars represent statistically significant correlations (p-value<0.10).

| | West Antarctica | East Antarctica | Antarctica |
|---|---|---|---|
| DA $\delta^{18}$O | -0.02 | -0.16 | -0.25 |
| DA SMB | -0.19 | 0.51 | 0.31 |
| DA $\delta^{18}$O and SMB | 0 | 0.60[*] | 0.44 |
| Stenni et al. (2017) | 0.45[*] | -0.20 | 0.12[*] |

**Table S3.** Slopes ($^{\circ}$C 100yr$^{-1}$) of each surface temperature reconstruction (Stenni et al., 2017; Klein et al., 2019; Nicolas and Bromwich, 2014; in this study) over the 1961–2010 period for West Antarctica, East Antarctica and the Antarctica. Statistically significant (p-value < 0.05) trends are represented by a star.

| Dataset | West Antarctica | East Antarctica | Antarctica |
|---|---|---|---|
| Stenni et al. (2017) | | | |
|     Stat ECHAMvariance | 1.69* | 0.75* | 1.27 |
|     Stat borehole | 2.07* | 0.75* | 0.77* |
| Klein et al. (2018) | | | |
|     DA ECHAM5-wiso | 1.15 | 0.94 | 0.98 |
|     DA ECHAM5/MPI-OM | 1.0 | 0.48 | 0.59 |
| Nicolas and Bromwich (2014) | | | |
| | 2.22* | 0.53 | 0.90* |
| In this study | | | |
|     DA $\delta^{18}$O and SMB iHadCM3 | 0.99* | 0.60* | 0.69* |

~~5-year mean correlations between the three surface temperature reconstructions from data assimilation experiments using the ECHAM5-MPI/OM outputs, ECHAM5-wiso outputs, the iHadCM3 outputs and the two surface temperature reconstructions of Stenni et al. (2017) with the surface temperature reconstruction from Nicolas and Bromwich (2014) over the 1958–2010 for East Antarctica, West Antarctica and Antarctica as a whole.~~

---

## Author Response (AR3)

Dear Editor,

Thank you again for editing our manuscript and for careful reading of the revised version that have allowed us to improve the clarity and correct some points.

You can find below the point by point response to each of your comments, along with the updated version of our manuscript with track changes.

On behalf of all co-authors,

Dalaiden Quentin

The Editor's comments below are in italics, our answer in plain font in blue.

*Dear authors,*

*The manuscript was revised adequately to address concerns brought by a referee. However, my final reading found many discrepancies between the numbers reported in Table 2 and numbers reported in the main text. I am afraid that the reviewers and the editor are not fully capable to check inter-consistency between these numbers, and we assume that the authors carefully checked these numbers. Please inform me if I read it wrong, but otherwise please give a very careful reading so that the paper is more accurate and consistent.*

We thank you for reading carefully the manuscript. You can find below our responses to each of your comments. We have also checked once more all the numbers used in the manuscript to avoid any mistake or misinterpretation.

*1. Define SAT = Surface air temperature in Fig. 3 caption or somewhere early enough in the main text. The current manuscript starts to use SAT from Fig. 3 caption but it is defined only in Fig. 5 caption. And I think "surface air temperature" and "surface temperature" are often used interchangeably. It is also said "near-surface temperature". Please give a careful final read to make sure that these relevant terms are used consistently and clearly. I think that the best way is to define SAT (like you define SMB) and keep using SAT.*

Now, we systematically use surface air temperature throughout the manuscript instead of "surface temperature" or "near-surface temperature". We also define SAT as surface air temperature at the beginning of the text.

*2. Table 1, define DA = Data Assimilation in caption.*

This is done.

*3. Table 2 should be cited much earlier. The table is located end of page 16 but it is first time cited at p19.*

We now cite Table 2 at the beginning of the section regarding our SMB reconstruction from data assimilation experiments (section 4.3.2). The reason why we cited this table later is explained in the next comment.

*4. Please check all numbers cited in Table 2 and text. For example, Table 2 shows that SMB trends over the entire Antarctica is 0.26 Gt/year2 for the period of 1801-2000 in the table, but the text says 0.33 Gt/year2 (P.16L15). There are similar discrepancies for other numbers reported in Table 2. Also check all reported numbers in the manuscript.*

Numbers in the text and in Table 2 were different because in section 4.3.2 - where we give our results regarding the SMB reconstruction - the trends are computed by using the AIS

mask including ice shelves. In Table 2, we compare our SMB reconstruction to the Medley and Thomas (2019) reconstruction, which is only available over the grounded AIS (i.e. the AIS excluding ice shelves). That is why numbers between the text and Table 2 were different.

We agree that those differences can be misleading. We thus suggest to only present trends computed over the grounded AIS, which allows us to directly compare our results with the results from Medley and Thomas (2019). The second paragraph of section 4.3.2 has been modified accordingly to stay consistent throughout the manuscript:

[revised manuscript text omitted]

**S1: Statistical surface air temperature reconstructions from Stenni et al. (2017)**

Using the $\delta^{18}O$ composites, Stenni et al. (2017) reconstructed regional surface air temperature over the last two millennia based on the statistical relationship between $\delta^{18}O$ and surface air temperature. Three methods have been used to scale the $\delta^{18}O$ composites. In the first approach, the regional slopes between $\delta^{18}O$ and temperatures were computed from the outputs of the ECHAM5-wiso model forced by ERA-Interim atmospheric reanalysis (Goursaud et al., 2018) over the 1979–2013 period. In the second method, the reconstruction obtained from the first method for the WAIS region is corrected using an independent temperature record: the borehole temperature reconstruction at WAIS divide (Orsi et al., 2012). This allows to match the cooling trend over the 1000–1600 period (Stenni et al., 2017). This method provides a different reconstruction for the WAIS region – implying thus also the West Antarctic and the whole Antarctic reconstructions –, but not for the regions in East Antarctica. Finally, in the third method, the regional normalized $\delta^{18}O$ composites have been scaled to the variance of the surface air temperature observations (e.g. Nicolas and Bromwich, 2014) over the 1960–1990 period. The second reconstruction is used throughout this study for two reasons: 1) the third method is based on the surface air temperature observations, which are used here to estimate the skill of the reconstructions which could lead to a bias; 2) the correction introduced in the second method is expected to improve the reconstruction compared to the previous method. The temporal resolution of these surface air temperature reconstruction is the same as the $\delta^{18}O$ composites: 10 years for 0–1800 period and 5 years for 1800–2010 period.

**S2: Defining uncertainties associated with proxy data used during data assimilation process**

Data assimilation requires estimates of the uncertainty associated with the proxy data used. Unfortunately, uncertainty estimations are not provided with published reconstructions used here and the instrumental time series are too short to reliably derive the uncertainty. If we apply the same error for all the Antarctic regions, the assimilation will tend to give more weight to the time series that have more variance (i.e. the high-accumulation regions). On the other hand, if we apply an error proportional to the standard deviation of the time series, each region will tend to have the same weight. The uncertainly could also be related to the amount of ice cores included in each regional composite, but the link between this number and the quality of the composite is not straightforward (Stenni et al., 2017). Several experiments have been performed to test the impact of different estimates of the data uncertainties on the data assimilation results. The results are qualitatively similar to standard choices of the uncertainty (Klein et al., 2019). The experiments shown here assume a signal to noise ratio of 1 for each regional composite. This is probably an optimistic estimate but this has the advantage of providing a strong data constraint and the comparison of the reconstruction using data assimilation with instrumental data indicates a good skill of the methods using this value.

**S3: Present-day AIS SMB simulated by GCMs**

Overall, the AIS SMB simulated by GCMs is in good agreement with the SMB simulated by the regional climate model RACMO2 over the last decades (1979–2005, $R^2$ = 0.63; Figs. S5, S6, S7 and S8). As expected, both display high values of

SMB along the coast (>300 mm w.e. year$^{-1}$) – especially for West Antarctica and the Antarctic Peninsula – and lower values at high elevations (e.g. the Plateau: <100 mm w.e. year$^{-1}$). The mean of the SMB over the entire AIS simulated by the selected models (including isotope-enable models) is 6.4 mm w.e. year$^{-1}$ lower than the SMB simulated by RACMO2 over the 1979–2005 period (relative bias: -3.4%; Fig. S8 for the correlation plots for each model and Fig. S9 for the integrated SMB over the entire AIS for each model).

However, Figure S5 shows that the GCMs, compared to RACMO2, underestimate SMB in areas below 1500 m (mean bias of -55 mm w.e. year$^{-1}$; relative bias: -15%) over 1979–2005. For the areas above 1500 m, the mean bias of the simulated SMB by GCMs compared to RACMO2 is 11 mm w.e. year$^{-1}$ (relative bias: 11%). These results are in agreement with previous studies (e.g. Palerme et al., 2017; Genthon et al., 2009; Krinner et al., 2008) who have shown that due to the lower spatial resolution of GCMs in comparison to the regional model, SMB is underestimated at the coasts while an overestimation occurs in the interior of the ice sheet. The bias in the difference between the coastal and higher elevation regions are smaller for the models that have a higher spatial resolution, such as CCSM4 (Fig. S10), confirming that the spatial resolution has a crucial impact on the simulated SMB. However, models with similar resolutions may also have very different results, in particular in coastal regions (relative SMB biases of +47% and +100% for CCSM4 and MRI-CGCM3 respectively compared to RACMO for DML coast over the 1979–2005 period), suggesting a critical role of model physics in some of the GCM biases.

[Figure]

**Figure S1.** Surface mass balance anomalies [Gt y$^{-1}$] simulated by the GCMs (the average of all the available simulations has been represented; Tab. A1) and snow accumulation reconstructions (Thomas et al., 2017) for 1000–2005 and for 1800–2005 for all the Antarctic subregions. Anomalies are computed for the 1800–2000 period. The shaded area corresponds to the range of the CESM1-CAM5 simulations. For visibility, data has been smoothed with a 100 year moving average for the last millennium and a 30 years moving average for the last 200 years.

[Figure]

**Figure S2.** Annual correlations (r) between surface mass balance and surface air temperature for all seven Antarctic regions (see Fig. 1 for geographical definitions) for all the GCMs over the 1850–2005 period.

[Figure]

**Figure S3.** 5-year mean correlations between surface air temperature and $\delta^{18}O$ (blue) and between SMB and surface air temperature (green) for the seven Antarctic regions for the entire period simulation (1871–2010 for ECHAM5-wiso and 801–2000 for ECHAM5/MPI-OM).

[Figure]

**Figure S4.** Spatial Antarctic surface mass balance trends (mm w.e. y$^{-1}$ decade$^{-1}$) over the 1801–2000, 1957–2000 and 1979–2000 periods from 1) our data assimilation-based reconstruction using the iHadCM3 outputs constrained by both $\delta^{18}$O and SMB (first row) and from 2) Medley and Thomas (2019; second row).

[Figure]

**Figure S5.** Antarctic Ice Sheet Surface Mass Balance [mm w.e. $y^{-1}$] over 1979–2005 CE averaged over all the GCMs simulations (see Tab. A1 for the list) (top left), for RACMO2 (van Wessem et al., 2018) (top right), the difference between them (bottom left) and the distribution of the SMB simulated by RACMO2 and the GCMs as a function of elevation, binned in 400m elevation intervals (bottom right). The bars represent one standard deviation of the cell grids within each elevation bin. The equivalent of the latter panel for each model is provided on Fig. S10.

[Figure]

**Figure S6.** Correlation plot of SMB climatology from GCM mean (average over all the GCMs including isotope-enabled models) as a function of RACMO SMB over the 1979–2005 period. $R^2$ is the determination coefficient and bias the average of the difference between GCM mean and RACMO (in mm w.e. year[-1]). Red (blue) dots are for places where the altitude is lower (higher) than 1500m. See Fig. S8 for each model.

[Figure]

**Figure S7.** Antarctic Ice Sheet surface mass balance [mm w.e. y$^{-1}$] for all the models used in this study over the 1979–2005 period.

[Figure]

**Figure S8.** As in Fig. S6 but for all GCMs.

[Figure]

**Figure S9.** Mean Antarctic Ice Sheet surface mass balance (Gt year$^{-1}$) simulated by all the models used in this study.

[Figure]

**Figure S10.** Distribution of the surface mass balance simulated by all climate models used in this study as a function of elevation, binned in 400m elevation intervals. The bars represent one standard deviation of the cell grids within each elevation bin.

[Figure]

**Figure S11.** Reconstructed surface air temperatures (5-year mean) for West Antarctica, East Antarctica and Antarctica as a whole from our data assimilation experiment using the ECHAM5-wiso outputs and, $\delta^{18}0$ (Stenni et al., 2017) and SMB reconstruction (Thomas et al., 2017) as data. The period is 1800–2010. *DA $\delta^{18}O$* is the data assimilation experiment using only the $\delta^{18}$O data to constrain the model while *DA SMB* uses only the SMB reconstruction and *DA $\delta^{18}O$ and SMB* uses both. For each experiment and each region, the correlation (*r*) between the reconstruction based on ice cores and that based on data assimilation is computed. The shaded areas represent $\pm$ 1 standard deviation of the model particles.

[Figure]

**Figure S12.** Reconstructed surface air temperatures (5-year mean) for West Antarctica, East Antarctica and Antarctica as a whole from data assimilation experiment using the ECHAM5-MPI/OM outputs and, $\delta^{18}$O (Stenni et al., 2017) and SMB reconstruction (Thomas et al., 2017) as data. The period is 1800–2010. *DA $\delta^{18}$O* is the data assimilation experiment using only the $\delta^{18}$O data to constrain the model while *DA SMB* uses only the SMB reconstruction and *DA $\delta^{18}$O and SMB* uses both. For each experiment and each region, the correlation (*r*) between the reconstruction based on ice cores and that based on data assimilation is computed. The shaded areas represent $\pm$ 1 standard deviation of the model particles.

[Figure]

**Figure S13.** Reconstructed SMB (5-year mean) for West Antarctica, East Antarctica and Antarctica as a whole from data assimilation experiment using the ECHAM5-wiso outputs and, $\delta^{18}O$ (Stenni et al., 2017) and SMB reconstruction (Thomas et al., 2017) as data. The period is 1800–2010. *DA $\delta^{18}O$* is the data assimilation experiment using only the $\delta^{18}O$ data to constrain the model while *DA SMB* uses only the SMB reconstruction and *DA $\delta^{18}O$ and SMB* uses both. For each experiment and each region, the correlation (*r*) between the reconstruction based on ice cores and that based on data assimilation is computed. The shaded areas represent ± 1 standard deviation of the model particles.

[Figure]

**Figure S14.** Reconstructed SMB (5-year mean) for West Antarctica, East Antarctica and Antarctica as a whole from data assimilation experiment using the ECHAM5-MPI/OM outputs and, $\delta^{18}$O (Stenni et al., 2017) and SMB reconstruction (Thomas et al., 2017) as data. The period is 1800–2010. *DA $\delta^{18}O$* is the data assimilation experiment using only the $\delta^{18}$O data to constrain the model while *DA SMB* uses only the SMB reconstruction and *DA $\delta^{18}O$ and SMB* uses both. For each experiment and each region, the correlation (*r*) between the reconstruction based on ice cores and that based on data assimilation is computed. The shaded areas represent $\pm$ 1 standard deviation of the model particles.

**Table S1.** Surface mass balance trends (in Gt 100y$^{-2}$) for West Antarctica, East Antarctica and Antarctica as a whole in GCMs, in isotopic climate models (ECHAM5-wiso, ECHAM5/MPIOM and HadCM3) and in reconstructions based on ice cores (Thomas et al., 2017) over 1950–2000. The number in brackets is the number of simulations. The trend computation is based on yearly data.

| | West Antarctica | | | East Antarctica | | | Antarctica | | |
|---|---|---|---|---|---|---|---|---|---|
| | min | max | mean | min | max | mean | min | max | mean |
| bcc-csm1-1 (3) | -29.46 | 152.47 | 77.62 | -63.11 | 381.09 | 200.39 | -92.57 | 533.56 | 278.01 |
| CCSM4 (6) | 148.02 | 390.50 | 234.13 | 274.32 | 455.65 | 368.19 | 469.24 | 846.15 | 602.32 |
| CSIRO-Mk3L-1-2 (1) | | | 3.14 | | | 135.86 | | | 139.00 |
| GISS-E2-R (6) | 25.69 | 183.66 | 107.27 | -71.92 | 250.18 | 140.43 | -46.23 | 416.21 | 247.71 |
| HadCM3 (10) | 4.79 | 150.27 | 70.75 | -68.85 | 242.18 | 89.39 | -34.18 | 303.11 | 160.14 |
| IPSL-CM5A-LR (6) | 57.07 | 123.78 | 99.07 | -104.18 | 66.82 | -10.06 | -47.11 | 174.30 | 89.01 |
| MPI-ESM-P (2) | -33.85 | -28.74 | -31.30 | 54.75 | 231.84 | 143.29 | 26.01 | 197.99 | 112.00 |
| MRI-CGCM3 (3) | 28.62 | 178.64 | 86.45 | -59.28 | 242.24 | 125.66 | -7.19 | 420.89 | 212.11 |
| CESM1-CAM5 (12) | 30.90 | 349.67 | 153.07 | 55.72 | 340.24 | 162.27 | 161.99 | 592.43 | 315.34 |
| iHadCM3 (6) | 76.23 | 232.69 | 162.29 | 15.52 | 350.87 | 213.61 | 115.85 | 542.61 | 375.90 |
| ECHAM5-wiso (1) | | | -8.79 | | | 195.22 | | | 186.43 |
| ECHAM5/MPIOM (1) | | | 41.44 | | | 35.43 | | | 76.87 |
| Reconstructions (1) | | | 256.74 | | | -35.80 | | | 220.95 |

**Table S2.** 5-year mean correlations between the three surface air temperature reconstructions from data assimilation experiments using the iHadCM3 outputs and the statistical reconstruction of Stenni et al. (2017), with the surface air temperature reconstructions from Nicolas and Bromwich (2014) over the 1958–2010 period for East Antarctica, West  Antarctica and Antarctica as a whole. All the correlations are performed on detrended time series. Stars represent statistically significant correlations (p-value<0.10).

| | West Antarctica | East Antarctica | Antarctica |
|---|---|---|---|
| DA $\delta^{18}$O | -0.02 | -0.16 | -0.25 |
| DA SMB | -0.19 | 0.51 | 0.31 |
| DA $\delta^{18}$O and SMB | 0 | 0.60[*] | 0.44 |
| Stenni et al. (2017) | 0.45[*] | -0.20 | 0.12[*] |

**Table S3.** Slopes (°C 100yr$^{-1}$) of each surface air temperature reconstruction (Stenni et al., 2017; Klein et al., 2019; Nicolas and Bromwich, 2014; in this study) over the 1961–2010 period for West Antarctica, East Antarctica and the Antarctica. Statistically significant (p-value < 0.05) trends are represented by a star.

| Dataset | West Antarctica | East Antarctica | Antarctica |
|---|---|---|---|
| Stenni et al. (2017) | | | |
| Stat ECHAMvariance | 1.69* | 0.75* | 1.27 |
| Stat borehole | 2.07* | 0.75* | 0.77* |
| Klein et al. (2018) | | | |
| DA ECHAM5-wiso | 1.15 | 0.94 | 0.98 |
| DA ECHAM5/MPI-OM | 1.0 | 0.48 | 0.59 |
| Nicolas and Bromwich (2014) | | | |
| | 2.22* | 0.53 | 0.90* |
| In this study | | | |
| DA $\delta^{18}$O and SMB iHadCM3 | 0.99* | 0.60* | 0.69* |

---

## Author Response (AR4)

Dear Editor,

We would like to thank you a lot for editing our manuscript as well as for its acceptance.

Best regards,

On behalf of all co-authors,

Dalaiden Quentin